# FedClean: A General Robust Label Noise Correction for Federated Learning

Xiaoqian Jiang [1]    Jing Zhang [1] *

## Abstract

Many federated learning scenarios encounter label noises in the client-side datasets. The resulting degradation in global model performance raises the urgent need to address label noise. This paper proposes FedClean – a novel general robust label noise correction for federated learning. FedClean first uses the local centralized noisy label learning to select clean samples to train a global model. Then, it employs a two-stage correction scheme to correct the noisy labels from two distinct perspectives of local noisy label learning and the global model. FedClean also proposes a novel model aggregation method, further reducing the impact of label noises. FedClean neither assumes the existence of clean clients nor the specific noise distributions, showing the maximum versatility. Extensive experimental results show that FedClean effectively identifies and rectifies label noises even if all clients exhibit label noises, which outperforms the state-of-the-art noise-label learning methods for federated learning.

## 1. Introduction

Federated learning (FL) has emerged as a powerful framework for decentralized machine learning (Liu et al., 2022), enabling multiple clients to collaboratively train models without sharing raw data, thus preserving privacy (Wen et al., 2023; Huang et al., 2023). This learning paradigm is especially valuable in privacy-sensitive domains (Nevrataki et al., 2023) such as healthcare (Rani et al., 2023; Thummisetti & Atluri, 2024), finance (Li & Wen, 2023; Awosika et al., 2024), and IoT (Yadav et al., 2022; Rjoub et al., 2024), where data privacy regulations like GDPR (Zaeem & Barber, 2020) are imposed to protect user information strictly. Since there is no centralized entity to preprocess the whole training data, the presence of label noise in client datasets

(Fang & Ye, 2022) becomes a critical challenge in federated learning, which significantly degrades the performance of global models (Zeng et al., 2023). Therefore, federated learning still has an urgent need to deal with label noise.

Label noise issue has been extensively studied within the context of centralized learning (CL), leading to the development of numerous advanced methods (Song et al., 2022). However, these CL-based methods cannot be directly applied to FL due to its inherent privacy constraints, which prohibit sharing data between clients or with a global server. As a result, the restricted size and insufficient diversity of the local datasets in clients dramatically degrade the effectiveness of noise processing methods. For example, Xu et al. (2022) have demonstrated that even the most sophisticated label noise correction methods for CL (Li et al., 2020a; Tanaka et al., 2018), when applied to client datasets, are insufficient to alleviate performance degradation in FL.

Therefore, recent studies have begun to address the label noise issue within the FL framework. Some methods attempt to mitigate label noise by discarding (Xu & Lyu, 2020) or re-weighting (Wan & Chen, 2021; Fu et al., 2021; Chen et al., 2020) the model updates in the clients that are least similar to those in other clients. They treat the discarded or re-weighted clients as malicious ones, which is obviously too radical. Many clients may simply have some label noises in their local datasets, which, with proper correction, could still contribute valuable information to the global model. To this end, latest methods were proposed to distinguish between *clean* (where all labels of local training data are correct) and *noisy* clients and then use models trained from clean clients to correct noisy labels in noisy clients (Xu et al., 2022; Wu et al., 2023; Jiang et al., 2024).

However, all existing label noise correction methods (Xu et al., 2022; Wu et al., 2023; Jiang et al., 2024) for FL are based on an idealized assumption that some clean clients exist, which unfortunately does not always hold in reality. For example, federated crowdsourcing learning (Guo et al., 2020) recruits non-expert workers from the Internet to collect data and train local models, where each client is probably imperfect. In other scenarios when clients are maliciously attacked, data are contaminated, or devices are compromised (Sharma & Marchang, 2024), the assumption also breaks up. Moreover, these methods distinguish

[1] School of Cyber Science and Engineering, Southeast University, No.2 SEU Road, Nanjing 211189, China. Correspondence to: Jing Zhang <jingz@seu.edu.cn>.

*Proceedings of the 42nd International Conference on Machine Learning*, Vancouver, Canada. PMLR 267, 2025. Copyright 2025 by the author(s).

between clean and noisy clients for different processing, potentially exposing client privacy. For instance, an attacker can infer from the communication patterns between clients and the server whether a client is noisy or clean (Bai et al., 2024). However, in real-world FL scenarios, clients are usually unwilling to disclose whether their data is noisy, as it may lower their reputation, or clean, as it may expose them to malicious attacks. What's more, some further assume specific noise distributions (class-conditional noise (Ji et al., 2024; Wu et al., 2023), instance-dependent noise (Wang et al., 2023)), further narrowing their application scope.

To address the challenges outlined above, we propose Fed-Clean, a general-purposed robust noise correction framework for FL. FedClean first uses the local centralized noisy label learning (CNLL) to select clean samples for global model training. Then, its two-stage correction scheme uses the information obtained from the local CNLL and the global model to correct the noisy labels. FedClean also proposes a novel adaptive sample size-weighted aggregation method (ASSA) to reduce the impact of label noise and further improve the performance of the global model. FedClean neither assumes the existence of clean clients nor the specific noise distributions. As such, FedClean shows outstanding robustness when the ratio of noisy clients is very high. Even if all clients are noisy, FedClean also performs well. The contributions of this study are three-fold:

1) We propose a novel general robust label noise correction for federated learning, where a two-stage label correction scheme is proposed to identify and rectify noisy labels from two distinct perspectives – local noisy label learning and global FL models. The framework also introduces a novel collaborative per-sample loss to assess the confidence in label corrections, which helps reduce the likelihood of false correction.

2) We propose a novel adaptive sample size-weighted aggregation (ASSA) method for FL that adjusts client influence based on clean sample sizes to reduce noisy label impact, while incorporating zkCor (Wang et al., 2024) for secure label correction.

3) Extensive experiments on datasets with synthetic label noises and a real-world dataset consistently demonstrate that the proposed FedClean effectively identifies and rectifies label noises hence mitigating the performance degradation of FL models, even if all the clients are affected by label noises, which outperforms the state-of-the-art noise-label learning methods for FL.

## 2. Related Work

### 2.1. Centralized Noisy Label Learning (CNLL)

In centralized learning, noisy label correction has been explored through various techniques. *Sample selection meth-*

*ods* like Co-teaching (Han et al., 2018) and Co-teaching+ (Yu et al., 2019) use two peer networks to exchange samples, with those having lower loss values assumed to be more reliable. *Robust loss functions* such as Symmetric CE (Wang et al., 2019) integrate model predictions into the loss function, while Joint Optim (Tanaka et al., 2018) refines labels by averaging model predictions over epochs. SELFIE (Song et al., 2019) focuses on robustly identifying potential noisy samples and gradually incorporating them into the training process. DivideMix (Li et al., 2020a) combines Co-teaching, MixUp (Zhang et al., 2018), and MixMatch (Berthelot et al., 2019) for more robust noise handling. These CNLL methods can be adapted to FL by integrating with standard aggregation techniques like FedAvg (McMahan et al., 2016).

### 2.2. Federated Noisy Label Learning (FNLL)

Label noise in FL has led to several strategies, primarily focusing on reweighting/discarding noisy data, leveraging clean datasets, and distinguishing between clean and noisy clients: i) *Reweighting and discarding noisy data.* Methods like Client Confidence Reweighting (CCR) (Fang & Ye, 2022) assign adaptive weights to clients based on data quality. FedNoiL (Wang et al., 2022) selects clients with fewer noisy labels and discards noisy samples, while RFFL (Xu & Lyu, 2020) uses reputation-based client evaluation to exclude unreliable clients. These methods may discard valuable data and fail to fully leverage noisy samples. ii) *Leveraging additional clean datasets.* Techniques like FO-CUS (Chen et al., 2020) use benchmark datasets on the server side to assess client data credibility and adjust client weights accordingly. Client selection algorithms (Yang et al., 2021) identify clients with low noise using clean validation datasets. However, these methods rely on the availability of clean datasets, which may not always be feasible. iii) *Distinguishing between clean and noisy clients.* More advanced methods dynamically distinguish between clean and noisy clients. FedCorr (Xu et al., 2022) uses model prediction subspaces to identify noisy clients, while FedNoRo (Wu et al., 2023) applies Gaussian mixture models and knowledge distillation for more robust aggregation. FedELC (Jiang et al., 2024) detects noisy clients and corrects their labels via backpropagation. However, these methods assume the existence of clean clients and struggle when such clients are absent or noisy labels are pervasive.

Other approaches to FNLL include Robust FL (Yang et al., 2022), which aligns client data via class-wise centroids, and FedLSR (Jiang et al., 2022), which employs self-distillation for local regularization to enhance privacy. FedRN (Kim et al., 2022) increases communication overhead by maintaining reliable neighbor models, while FedFixer (Ji et al., 2024) uses personalized models to mitigate the impact of noisy labels but is dependent on specific noise distributions.

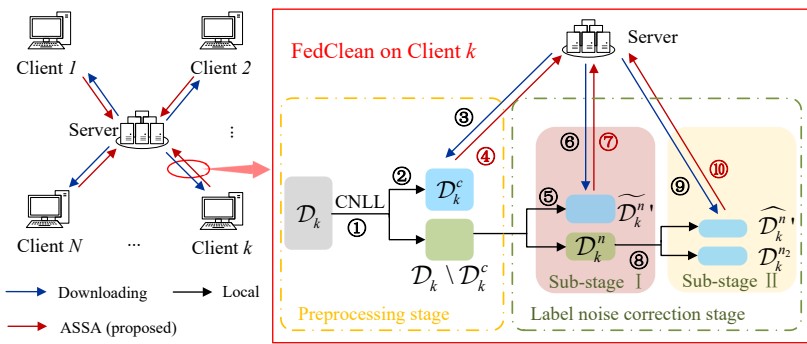

*Figure 1.* Framework of FedClean. Algorithm steps are numbered accordingly.

# 3. The Proposed Method

## 3.1. Preliminaries

We consider a federated learning system consisting of $N$ clients $\mathcal{N} = \{1, ..., N\}$ and their local dataset $\{\mathcal{D}_k\}_{k=1}^{N}$, where $\mathcal{D}_k = \{(x_k^i, y_k^i)\}_{i=1}^{n_k}$ denotes the local dataset for client $k \in \mathcal{N}$, $y_k^i$ represents the annotation label for the sample $x_k^i$. At the conclusion of the $t$-round communication, $w_k^t$ is defined as the local model weight of client $k$, while $w^t$ represents the weight of the aggregated global model $\theta_G^t$. And $\mathcal{N}^t \subseteq \mathcal{N}$ is the subset of selected clients in round $t$. This process of federated training closely resembles that of the usual FL (FedAvg), with two key differences: the integration of the Mixup technique and adaptive sample size-weighted aggregation.

*Mixup.* Mixup (Zhang et al., 2018) is a data augmentation technique exhibiting strong robustness to label noise. Given two samples $(x_k^i, y_k^i) \in$ and $(x_k^j, y_k^j)$ in dataset $\mathcal{D}_k^c$, it generates a new sample $(\tilde{x}_k, \tilde{y}_k)$ by linearly combining the two original samples with a random weight. Specifically, the new sample $\tilde{x}_k$ and its corresponding label $\tilde{y}_k$ are given by:

$$\tilde{x}_k = \lambda \cdot x_k^i + (1 - \lambda) \cdot x_k^j, \tag{1}$$

$$\tilde{y}_k = \lambda \cdot y_k^i + (1 - \lambda) \cdot y_k^j, \tag{2}$$

where $\lambda$ is a random hyperparameter drawn from the Beta distribution $\beta(\alpha, \alpha)$, and $\alpha$ is a parameter that controls the shape of the distribution. In our experiments, we set $\alpha = 1$. Mixup effectively mitigates the negative impact of noisy labels by smoothing the labels and is robust to label noise. Suppose $y_k^i$ represents a noisy label and $y_k^j$ is the true label. According to the Mixup formulation Eq (2), the new label $\tilde{y}_k$ is a weighted combination of the two original labels $y_k^i$ and $y_k^j$. Specifically, when $\lambda$ is small, $\tilde{y}_k$ will be closer to the true label $y_k^j$, thus diminishing the influence of the noisy label $y_k^i$. This label smoothing property of Mixup reduces the negative effects of label noise, making the model more robust to such noise.

## 3.2. Framework of FedClean

We propose FedClean, a general robust label noise correction method for FL. The framework of FedClean is shown in Figure 1, which has two key stages. In the preprocessing stage, FedClean enables each client to perform local CNLL, generating inferred labels for each sample and initially screening clean samples to train a global model. During the label noise correction stage, we propose a two-stage correction scheme that integrates inferred labeling with the global model to accurately identify and correct noisy labels. Within this inferred label-based correction process, we define a collaborative per-sample loss function to evaluate the confidence of the label corrections. To enhance the robustness of federated training against label noise, we incorporate the Mixup and the proposed Adaptive Sample Size-weighted Aggregation (ASSA). Notably, FedClean does not distinguish between "clean" and "noisy" clients, effectively correcting label noise in client data while adhering to the privacy constraints of FL.

*Adaptive Sample Size-weighted Aggregation (ASSA).* The proposed ASSA appears at steps ④, ⑦ and ⑩ in Figure 1, where the weight coefficients for each client in the global model update are determined by the number of samples actually participating in the federated training, rather than being fixed based on the size of the client's entire dataset. This approach ensures that only the samples actively involved in the federated learning process influence the global model's weight update. ASSA demonstrates robustness to label noise, particularly for clients with high levels of label noise. The details of the weight update steps are presented in the subsequent sections.

## 3.3. Preprocessing Stage

During the preprocessing stage, we utilize the CNLL to identify clean samples for each client. Only the selected clean samples are subsequently used in the federated training process of this stage.

**Add inferred labels.** We first let each client $k$ execute CNLL locally to train local model $\theta_k$. Subsequently, each sample $x_k^i \in \mathcal{D}_k$ is assigned an additional inferred label $\bar{y}_k^i$ based on the predictions of this local model. As a result, in addition to the annotation label $y_k^i$, each sample $x_k^i$ is also assigned an inferred label $\bar{y}_k^i$. At this stage, we represent the local dataset of each client as $\mathcal{D}_k = \{(x_k^i, y_k^i, \bar{y}_k^i) | \bar{y}_k^i = \theta_k(x_k^i)\}$.

**Select clean samples.** We select the samples whose annotation labels and inferred labels are identical. These samples are considered clean and denoted by:

$$\mathcal{D}_k^c = \{(x_k^i, y_k^i, \bar{y}_k^i) \in \mathcal{D}_k | y_k^i = \bar{y}_k^i\}. \qquad (3)$$

The remaining samples in $\mathcal{D}_k \setminus \mathcal{D}_k^c$ are deemed disputed samples. Note that, in the process of selecting clean samples, even if a sample with a noisy annotation label is mistakenly included in the clean set ($\mathcal{D}_k^c$) due to the wrong prediction by the local model $\theta_k$ (i.e., $\theta_k$'s predicted label of the example is the same as its noisy annotation label), the impact of this noisy label on the global model can be mitigated by the application of Mixup.

**ASSA.** Finally, we train the global model over $T_1$ rounds using these selected clean samples $\{\mathcal{D}_k^c\}_{k=1}^N$ on all clients. The weighting coefficients in ASSA at this stage are determined by the size of the clean dataset $\mathcal{D}_k^c$ selected by each client. The specific weight update is as follows:

$$w^t \leftarrow \sum_{k \in \mathcal{N}^t} \frac{|\mathcal{D}_k^c|}{\sum_{i \in \mathcal{N}^t} |\mathcal{D}_i^c|} \cdot w_k^t. \qquad (4)$$

Applying ASSA at this stage exhibits strong robustness, particularly for clients with high levels of label noise. Specifically, consider a client $k$ that experiences a significant amount of label noise, resulting in a relatively small clean sample set ($\mathcal{D}_k^c$). In ASSA, the weight of each client's local model in the global model update is proportional to the size of its clean sample set. Consequently, the influence of clients with smaller clean sample sets on the global model is correspondingly reduced. In other words, even if the local model trained by the client on its limited clean samples contains some degree of error, its contribution to the global model's weight will be relatively minor (namely $\frac{|\mathcal{D}_k^c|}{\sum_{i \in \mathcal{N}^t} |\mathcal{D}_i^c|} \ll 1$), thereby mitigating the negative impact of label noise on the global model.

### 3.4. Label Noise Correction Stage

To improve the accuracy of label correction and prevent overcorrection, we propose a two-stage label noise correction scheme. It consists of two sub-stages that are dominated by inferred labels and by the global model, respectively.

**SUB-STAGE I**: CORRECTION DOMINATED BY INFERRED LABELS. In this sub-stage, the limited number of clean samples selected during the preprocessing stage results in insufficient accuracy of the trained global model. To address this, we introduce inferred labels as supplementary information.

**Theorem 3.1.** *Incorporating inferred labels $\bar{y}_k^i$ as prior information refines the model's predictions, leading to a performance improvement bounded as:*

$$\mathcal{A}(\hat{y}_k^i, \bar{y}_k^i) \geq \mathcal{A}(\hat{y}_k^i) + \delta, \qquad (5)$$

*where $\delta$ quantifies the reduction in prediction error after combining inferred labels and $\mathcal{A}$ indicates the prediction accuracy.*

*Proof.* For sample $(x_k^i, y_k^i, \bar{y}_k^i)$, Let $\hat{y}_k^i$ be the global model $\theta_G^{T_1}$'s prediction and $y_k^i(\text{true})$ be the true label. Using Bayes' theorem, we update the global model's prediction as:

$$P(\hat{y}_k^i | \bar{y}_k^i) = \frac{P(\bar{y}_k^i | \hat{y}_k^i) \cdot P(\hat{y}_k^i)}{P(\bar{y}_k^i)}, \qquad (6)$$

This shows how incorporating the inferred label $\bar{y}_k^i$ as prior information refines the global model's prediction. We refine the predictions using the inferred labels as priors, expressed through the MAP (maximum a posteriori) estimate:

$$\hat{y}_k^i(\bar{y}_k^i) = \arg\max_{\hat{y}_k^i} P(\hat{y}_k^i | \bar{y}_k^i), \qquad (7)$$

Then, we can define the error of the global model as:

$$\epsilon = \frac{1}{S} \sum_{i=1}^S \mathbb{I}(\hat{y}_k^i \neq y_k^i(\text{true})), \qquad (8)$$

where $\mathbb{I}(\cdot)$ is the indicator function and $S$ is the number of samples. After incorporating the inferred labels as priors, the error becomes:

$$\epsilon' = \frac{1}{S} \sum_{i=1}^S \mathbb{I}(\hat{y}_k^i(\bar{y}_k^i) \neq y_k^i(\text{true})), \qquad (9)$$

Therefore, using the inference label as "prior information" improves the accuracy $\mathcal{A}$ of the model as follows:

$$\mathcal{A}(\hat{y}_k^i, \bar{y}_k^i) \geq \mathcal{A}(\hat{y}_k^i) + \delta, \quad \delta = \epsilon - \epsilon'. \qquad (10)$$

To rigorously quantify the reduction in prediction errors, we provide a detailed proof in Appendix B, linking the error reduction to the Kullback-Leibler divergence between the model's prior and posterior distributions. $\square$

For controversial samples (samples whose the annotation labels conflict with their inferred labels), we propose a novel collaborative per-sample loss:

$$\mathcal{L}_{\text{co}}(y_k^i, \bar{y}_k^i, \hat{y}_k^i; \theta_G^{T_1}) = \mathcal{L}_{\text{an}}(y_k^i, \hat{y}_k^i; \theta_G^{T_1}) - \mathcal{L}_{\text{in}}(\bar{y}_k^i, \hat{y}_k^i; \theta_G^{T_1}). \qquad (11)$$

Here, $\mathcal{L}_{\mathrm{an}}(y_k^i, \hat{y}_k^i; \theta_G^{T_1})$ represents the annotation label loss, measuring the discrepancy between the annotation label of sample $x_k^i$ and the model's prediction. This loss is typically larger if there is noise in the annotation label. $\mathcal{L}_{\mathrm{in}}(\bar{y}_k^i, \hat{y}_k^i; \theta_G^{T_1})$ denotes the inferred label loss, quantifying the misalignment between the sample's annotation and its assigned inferred label. When the inferred label is closer to the true label or more consistent, this loss is smaller.

Consequently, collaborative per-sample loss serves two main purposes. On the one hand, the collaborative per-sample loss can be interpreted as a measure of the consistency and confidence between the two labels (annotation label vs. inferred label). If a sample's annotation and inferred labels are consistent, i.e., $\mathcal{L}_{\mathrm{an}}(y_k^i, \hat{y}_k^i; \theta_G^{T_1}) = \mathcal{L}_{\mathrm{in}}(\bar{y}_k^i, \hat{y}_k^i; \theta_G^{T_1})$, then $\mathcal{L}_{\mathrm{co}}(y_k^i, \bar{y}_k^i, \hat{y}_k^i; \theta_G^{T_1}) = 0$, implying that the sample has minimal label noise and both labels are reliable. This provides a theoretical basis for classifying samples with matching annotation and inferred labels as clean samples during the preprocessing stage. On the other hand, collaborative per-sample loss also quantifies the degree of inconsistency between the annotation label and the inferred label. When the annotation label is erroneous (i.e., contains noise), $\mathcal{L}_{\mathrm{an}}(y_k^i, \hat{y}_k^i; \theta_G^{T_1})$ will be larger while $\mathcal{L}_{\mathrm{in}}(\bar{y}_k^i, \hat{y}_k^i; \theta_G^{T_1})$ remains relatively smaller, leading to a larger $\mathcal{L}_{\mathrm{co}}(y_k^i, \bar{y}_k^i, \hat{y}_k^i; \theta_G^{T_1})$. In conclusion, The greater the collaborative per-sample loss ($\mathcal{L}_{\mathrm{co}}(y_k^i, \bar{y}_k^i, \hat{y}_k^i; \theta_G^{T_1}) > 0$), the larger the discrepancy between the annotation and inferred labels, indicating that the annotation label is likely incorrect and can be corrected by the inferred label. We provide further explanation in Appendix A for choosing per-sample loss over using labeled per-sample loss alone.

**Theorem 3.2.** *In collaborative per-sample losses, inferred labels provide additional stability, allowing for more accurate estimates of true labels.*

*proof.* Bayes' theorem allows us to compute the posterior probability of the annotation label $y_k^i$ given the model's prediction $\hat{y}_k^i$ and the inferred label $\bar{y}_k^i$. Therefore, the posterior probability of Eq. (11) can be written as:

$$P(y_k^i | \hat{y}_k^i, \bar{y}_k^i) \propto P(\hat{y}_k^i | y_k^i) P(\bar{y}_k^i | y_k^i) P(y_k^i). \quad (12)$$

For correct annotations, both $P(\hat{y}_k^i | y_k^i)$ and $P(\bar{y}_k^i | y_k^i)$ will be high. For noisy annotations, $P(\hat{y}_k^i | y_k^i)$ will be low, but $P(\bar{y}_k^i | y_k^i)$ will still provide a stable signal, allowing us to better estimate the true label. This shows that the inferred label $\bar{y}_k^i$ provides additional stability when the annotation label is noisy, allowing for a more accurate estimation of the true label. (For more detailed stability analysis and proof, see Appendix C.) $\square$

*Validity of model predictions consistent with inferred labels.* It is important to note that correcting annotation labels based on collaborative per-sample loss is only meaningful if the

model's prediction is consistent with the inferred label. This is because: i) If the model's predicted label aligns with the inferred label, it indicates that the model has effectively learned the pattern of the inferred label. The inferred label likely represents the true class of the sample or is closer to it. ii) If the collaborative per-sample loss is large under these conditions, it indicates a significant mismatch between the annotation and inferred labels, suggesting that the annotation label is likely incorrect and should be corrected by the inferred label. Conversely, if the model's prediction conflicts with the inferred label, then the inferred label may not be reliable, and correcting the annotation label based on it could lead to errors. Thus, the collaborative per-sample loss should only be used to guide label correction when the model's prediction aligns with the inferred label. The specific steps for Sub-stage I are as follows:

**Calculate the collaborative per-sample loss.** To optimize computational efficiency, we calculate collaborative per-sample losses only for those samples where the global model's prediction matches the inferred label. For each client, these samples are recorded as the set:

$$\widetilde{\mathcal{D}_k^n} = \{(x_k^i, y_k^i, \bar{y}_k^i) \in \mathcal{D}_k \setminus \mathcal{D}_k^c | \bar{y}_k^i = \hat{y}_k^i\}. \quad (13)$$

Then, each client locally computes a Gaussian Mixture Model (GMM) on the collaborative per-sample loss values for all samples in the set $\widetilde{\mathcal{D}_k^n}$ to partition the set into two subsets: a subset $\widetilde{\mathcal{D}_k^{n_1}}$ that can be corrected by inferred labels and a disputed subset $\widetilde{\mathcal{D}_k^{n_2}}$.

**Filter samples for correction.** To avoid overcorrection, we apply the correction only to those samples identified as having a high-confidence inferred label. We introduce a correction rate $\sigma_1$, where for client $k$, we select the top $\sigma_1$-percent of samples from $\widetilde{\mathcal{D}_k^{n_1}}$ that have the highest collaborative per-sample loss. These samples are then relabeled using the inferred label $\bar{y}_k^i$. The subset of samples to be relabeled is denoted by:

$$\widetilde{\mathcal{D}_k^n}' = \arg \max_{\substack{\tilde{\mathcal{D}} \subseteq \widetilde{\mathcal{D}_k^{n_1}} \\ |\tilde{\mathcal{D}}| = \sigma_1 \cdot |\widetilde{\mathcal{D}_k^{n_1}}|}} \mathcal{L}_{\mathrm{co}}(\tilde{\mathcal{D}}; \theta_G^{T_1}). \quad (14)$$

**ASSA.** Finally, the global model is improved using the corrected samples by updating the parameter over $T_2$ rounds. In this stage of federated training, the global model $\theta_G^{T_1}$ updates its weights $w^t$ using ASSA as follows:

$$w^t \leftarrow \sum_{k \in \mathcal{N}^t} \frac{|\mathcal{D}_k^c| + |\widetilde{\mathcal{D}_k^n}'|}{\sum_{i \in \mathcal{N}^t}(|\mathcal{D}_i^c| + |\widetilde{\mathcal{D}_k^n}'|)} \cdot w_k^t. \quad (15)$$

The weight $\sum_{k \in \mathcal{N}^t} \frac{|\mathcal{D}_k^c| + |\widetilde{\mathcal{D}_k^n}'|}{\sum_{i \in \mathcal{N}^t}(|\mathcal{D}_i^c| + |\widetilde{\mathcal{D}_k^n}'|)}$ represents the cumulative impact of client $k$ on the global model, thereby further

**Algorithm 1** FedClean

---

1: **Input:** $N$ (number of clients), $T_1, T_2, T_3$ (number of rounds of communication), $\mathcal{D} = \{\mathcal{D}_k\}_{k=1}^N$ (dataset), $\theta_G^0$ ((initialized global model).
2: **Output:** Global model $\theta_G$.
  *//Preprocessing stage.*
3: **for** $k = 1$ to $N$ **do**
4:   Train model $\theta_k$ on $\mathcal{D}_k$ by CNLL;
5:   Add the inferred label $\bar{y}_k^j = \theta_k(x_k^i)$ to sample $x_k^i$, $\forall(x_k^i, y_k^i) \in \mathcal{D}_k$;
6:   Select clean samples via Eq. (3);
7: **end for**
8: **for** $t = 1$ to $T_1$ **do**
9:   Train global model $\theta_G^t$ on $\{\mathcal{D}_k^c\}_{k=1}^N$ via Eq. (4);
10: **end for**
  *//Label correction stage.*
  *//Sub-stage I: Correction stage dominated by inferred labels.*
11: **for** $k = 1$ to $N$ **do**
12:   Put $(x_k^i, y_k^i, \bar{y}_k^i)$ into the set $\widetilde{\mathcal{D}_k^n}$ via Eq. (13);
13:   Calculate $\mathcal{L}_{co}(y_k^i, \bar{y}_k^i, \hat{y}_k^i; \theta_G^{T_1})$ via Eq. (11), $\forall(x_k^i, y_k^i, \bar{y}_k^i) \in \widetilde{\mathcal{D}_k^n}$;
14:   Divide $\widetilde{\mathcal{D}_k^n}$ into $\widetilde{\mathcal{D}_k^{n_1}}$ and $\widetilde{\mathcal{D}_k^{n_2}}$ based on $\mathcal{L}_{co}(y_k^i, \bar{y}_k^i, \hat{y}_k^i; \theta_G^{T_1})$ via GMM;
15:   $y_k^i \rightarrow \bar{y}_k^i, \forall(x_k^i, y_k^i, \bar{y}_k^i) \in \widetilde{\mathcal{D}_k^n}'$ via Eq. (14);
16: **end for**
17: **for** $t = T_1 + 1$ to $T_1 + T_2$ **do**
18:   Train global model $\theta_G^t$ on $\{\widetilde{\mathcal{D}_k^n}'\}_{k=1}^N$ via Eq. (15);
19: **end for**
  *//Sub-stage II: Correction stage dominated by global model.*
20: **for** $k = 1$ to $N$ **do**
21:   Calculate $\mathcal{L}_{an}(y_k^i, \hat{y}_k^i; \theta_G^{T_1+T_2})$, $\forall(x_k^i, y_k^i) \in \mathcal{D}_k^n = \{(x_k^i, y_k^i)|(x_k^i, y_k^i, \bar{y}_k^i) \in \mathcal{D}_k \setminus (\mathcal{D}_k^s \cup \widetilde{\mathcal{D}_k^n}')\}$;
22:   Divide $\mathcal{D}_k^n$ into $\mathcal{D}_k^{n_1}$ and $\mathcal{D}_k^{n_2}$ based on $\mathcal{L}_{an}(y_k^i, \hat{y}_k^i; \theta_G^{T_1+T_2})$ via GMM;
23:   Put $(x_k^i, y_k^i)$ into the set $\widehat{\mathcal{D}_k^n}$ via Eq. (16);
24:   $y_k^i \rightarrow \hat{y}_k^i, \forall(x_k^i, y_k^i) \in \widehat{\mathcal{D}_k^n}$ via Eq. (17);
25: **end for**
26: **for** $t = T_1 + T_2 + 1$ to $T_1 + T_2 + T_3$ **do**
27:   Train global model $\theta_G^t$ on $\{\widehat{\mathcal{D}_k^n}'\}_{k=1}^N$ via Eq. (18);
28: **end for**
29: **Return** $\theta_G^{T_1+T_2+T_3}$.

---

mitigating the negative effects of noise correction errors at this stage. For client $k$, if $\widetilde{\mathcal{D}_k^n}' = \emptyset$, then $w_k^t \leftarrow w_k^{T_1}$.

**SUB-STAGE II**: CORRECTION DOMINATED BY GLOBAL LABELS. In the first sub-stage, we perform preliminary label corrections using a strategy led by the inferred label and assisted by the global model, which improves the accuracy of the global model. However, since the global model may not be fully optimized during the initial training phase, there may still be a small number of samples that are not effectively corrected. To address this, in the second sub-stage, we employ a global model-led label correction method to further enhance label accuracy. The primary objective of this sub-stage is to refine the sample labels using the globally trained model. Specifically, we calculate the loss for

each sample using the global model, identify those with higher loss values, and replace their annotation labels with the model's predicted labels, thereby correcting the labels. The specific steps for Sub-stage II are as follows:

**Calculate the per-sample loss.** We first compute the per-sample loss for each sample in the remaining dataset $\mathcal{D}_k^n = \{(x_k^i, y_k^i)|(x_k^i, y_k^i, \bar{y}_k^i) \in \mathcal{D}_k \setminus (\mathcal{D}_k^c \cup \widetilde{\mathcal{D}_k^n}')\}$, where the loss for each sample, denoted by $\mathcal{L}_{an}(\mathcal{D}_k^n; \theta_G^{T_1+T_2})$, represents the discrepancy between the model $\theta_G^{T_1+T_2}$'s prediction and the annotation label. The aim is to correct the noisy annotation labels, and thus the loss is calculated relative to the annotation label. Then, each client locally computes a GMM on the per-sample loss values for all samples in the set $\mathcal{D}_k^n$ to partition the set into two subsets: a noisy subset $\mathcal{D}_k^{n_1}$ and a clean subset $\mathcal{D}_k^{n_2}$.

*Table 1.* List of datasets used in our experiments.

| Dataset | CIFAR-10 | CIFAR-100 | Clothing1M |
|---|---|---|---|
| Size | 50000 | 50000 | 1000000 |
| # Classes | 10 | 100 | 14 |
| # Clients | 50 | 50 | 300 |
| # Rounds $T_1/T_2/T_3$ | 100/150/150 | 100/150/150 | 50/100/100 |
| Learning rate | 0.01 | 0.01 | 0.03 |
| Batch size | 10 | 10 | 20 |
| Architecture | ResNet-18 | ResNet-34 | ResNet-50 |

**Select samples for correction.** For all samples in noisy subset $\mathcal{D}_k^{n_1}$, we identify and select only those samples with high loss values, which are likely to contain label noise. To prevent overcorrection, we introduce two parameters – the correction rate $\sigma_2$ and the confidence threshold $\varepsilon$. (At this correction sub-stage, the only reference available is the global model, which is not fully reliable. To enhance its reliability, a confidence threshold is applied. In the sub-stage I, noise labels are corrected using both the inferred label and the inferred model, which has proven reliable enough in theorem 3.1 to not require an additional confidence threshold.) Specifically, for client $k$, we first identify the top $\sigma_2$-percent of samples from $\mathcal{D}_k^{n_1}$ that have the highest collaborative per-sample loss, denoted by:

$$\widehat{\mathcal{D}_k^n} = \arg \max_{\substack{\hat{\mathcal{D}} \subseteq \mathcal{D}_k^{n_1} \\ |\hat{\mathcal{D}}| = \sigma_2 \cdot |\mathcal{D}_k^{n_1}|}} \mathcal{L}_{an}(\hat{\mathcal{D}}; \theta_G^{T_1+T_2}). \quad (16)$$

Next, we compute the prediction vector $\theta_{G_2}(\widehat{\mathcal{D}_k^n}')$ from the global model and select the sample for re-labeling only if the maximum value in $\theta_{G_2}(\widehat{\mathcal{D}_k^n})$ exceeds the confidence threshold $\varepsilon$. Thus, the subset of samples to be re-labeled is denoted by:

$$\widehat{\mathcal{D}_k^n}' = \{(x_k^i, y_k^i) \in \widehat{\mathcal{D}_k^n} | \max(\theta_G^{T_1+T_2}(x_k^i)) \geq \varepsilon\}. \quad (17)$$

Finally, we use the global model $\theta_G^{T_1+T_2}$'s predicted labels ($\hat{y}_k^i$s) to correct the annotation labels ($y_k^i$s) of the samples in the subset $\widehat{\mathcal{D}_k^n}'$.

*Table 2.* Average (5 trials) accuracies (%) of various methods on CIFAR-10 dataset with IID and non-IID settings at different noise levels ($\rho$: ratio of noisy clients, $\tau$: lower bound of client noise level). The best results are highlighted in bold.

| Methods | IID | | | non-IID | | |
|---|---|---|---|---|---|---|
| | $\rho = 0$ $\tau = 0$ | $\rho = 0.5$ $\tau = 0.3$ | $\rho = 1$ $\tau = 0.5$ | $\rho = 0$ $\tau = 0$ | $\rho = 0.5$ $\tau = 0.3$ | $\rho = 1$ $\tau = 0.5$ |
| FedAvg | $91.74 \pm 0.19$ | $83.16 \pm 0.31$ | $38.36 \pm 2.21$ | $90.04 \pm 0.17$ | $82.61 \pm 0.26$ | $34.65 \pm 1.53$ |
| FedProx | $91.52 \pm 0.22$ | $82.45 \pm 0.27$ | $35.21 \pm 1.75$ | $\mathbf{90.82 \pm 0.18}$ | $81.76 \pm 0.22$ | $32.84 \pm 1.65$ |
| FedCorr | $\mathbf{91.83 \pm 0.21}$ | $\mathbf{91.12 \pm 0.30}$ | $47.49 \pm 1.98$ | $90.21 \pm 0.16$ | $\mathbf{89.11 \pm 0.25}$ | $39.40 \pm 1.51$ |
| FedNoRo | $90.05 \pm 0.19$ | $88.48 \pm 0.24$ | $32.18 \pm 1.89$ | $88.91 \pm 0.20$ | $86.99 \pm 0.21$ | $30.21 \pm 1.72$ |
| FedBeat | $89.28 \pm 0.23$ | $85.92 \pm 0.28$ | $36.13 \pm 2.03$ | $89.55 \pm 0.19$ | $83.92 \pm 0.24$ | $33.20 \pm 1.62$ |
| FedELC | $85.62 \pm 0.20$ | $87.60 \pm 0.29$ | $35.72 \pm 2.10$ | $89.90 \pm 0.17$ | $83.75 \pm 0.23$ | $31.95 \pm 1.80$ |
| FedFixer | $90.72 \pm 0.47$ | $87.06 \pm 0.30$ | $62.87 \pm 0.17$ | $89.76 \pm 0.32$ | $87.82 \pm 0.22$ | $59.01 \pm 0.55$ |
| FedClean[1] | $88.77 \pm 0.17$ | $85.25 \pm 0.29$ | $81.68 \pm 2.10$ | $87.79 \pm 0.20$ | $86.53 \pm 0.25$ | $77.12 \pm 1.95$ |
| FedClean[2] | $91.14 \pm 0.19$ | $88.41 \pm 0.27$ | $\mathbf{83.75 \pm 2.03}$ | $89.34 \pm 0.21$ | $86.79 \pm 0.23$ | $\mathbf{80.55 \pm 1.82}$ |

**ASSA.** After correcting the annotation labels in the second stage, we train the global model over $T_3$ rounds using the these corrected labels and the clean samples in $\mathcal{D}_k^{n_2}$. We continue to leverage the cumulative impact of clients on the global model in ASSA as follows:

$$w^t \leftarrow \sum_{k \in \mathcal{N}^t} \frac{|\mathcal{D}_k^c| + |\widetilde{\mathcal{D}_k^n}'| + |\widehat{\mathcal{D}_k^n}'| + |\mathcal{D}_k^{n_2}|}{\sum_{i \in \mathcal{N}^t}(|\mathcal{D}_i^c| + |\widetilde{\mathcal{D}_k^n}'| + |\widehat{\mathcal{D}_k^n}'| + |\mathcal{D}_k^{n_2}|)} \cdot w_k^t. \quad (18)$$

For client $k$, if $\widehat{\mathcal{D}_k^n}' \cup \mathcal{D}_k^{n_2} = \emptyset$, then $w_k^t \leftarrow w_k^{T_1 + T_2}$.

### 3.5. Algorithm

The pseudo-code of the FedClean method is shown in Algorithm 1, which is easily mapped into the technical content in preprocessing stage (lines 3-10) and label correction stage (Sub-stage I: lines 11-19. Sub-stage II: lines 20-29).

## 4. Experiments

We first conduct experiments on benchmark datasets with varying label noise settings to make comprehensive comparisons with state-of-the-art methods. Then, we design ablation studies to show the features of FedClean.

### 4.1. Experimental Setup

*Methods in comparison.* We compared the proposed FedClean method with two baseline approaches, FedAvg (McMahan et al., 2016), FedProx (Li et al., 2020b), as well as five state-of-the-art methods: FedCorr (Xu et al., 2022), FedNoRo (Wu et al., 2023), FedBeat (Wang et al., 2023), FedFixer (Ji et al., 2024), FedELC (Jiang et al., 2024). For the CNLL employed in FedClean, we selected one representative method from each of the two key approaches discussed in Subsection 2.1: Co-teaching (Han et al., 2018) (FedClean[1]) and Joint Optim (Tanaka et al., 2018) (FedClean[2]).

*Implementation details.* We evaluated different approaches under both IID (Independent and Identically Distributed) (CIFAR-10/100 (Krizhevsky, 2009)) and non-IID (non-Independent and Identically Distributed (Ma et al., 2022)) (CIFAR-10, Clothing1M (Xiao et al., 2015)) data settings.

*Table 3.* Average (5 trials) accuracies (%) of various methods on CIFAR-100 dataset with IID setting at different noise levels ($\rho$: ratio of noisy clients, $\tau$: lower bound of client noise level). The best results are highlighted in bold.

| Methods | $\rho = 0$ $\tau = 0$ | $\rho = 0.5$ $\tau = 0.3$ | $\rho = 1$ $\tau = 0.5$ |
|---|---|---|---|
| FedAvg | $\mathbf{72.36 \pm 0.19}$ | $62.12 \pm 0.25$ | $31.34 \pm 0.91$ |
| FedProx | $72.04 \pm 0.12$ | $63.53 \pm 0.20$ | $32.51 \pm 0.88$ |
| FedCorr | $72.33 \pm 0.16$ | $\mathbf{72.40 \pm 0.19}$ | $40.92 \pm 0.79$ |
| FedNoRo | $71.78 \pm 0.22$ | $67.02 \pm 0.24$ | $38.12 \pm 0.85$ |
| FedBeat | $70.82 \pm 0.28$ | $68.01 \pm 0.26$ | $30.74 \pm 0.92$ |
| FedELC | $71.82 \pm 0.18$ | $70.16 \pm 0.22$ | $31.45 \pm 0.93$ |
| FedClean[1] | $69.86 \pm 0.33$ | $68.75 \pm 0.21$ | $63.11 \pm 0.92$ |
| FedClean[2] | $70.94 \pm 0.25$ | $71.20 \pm 0.18$ | $\mathbf{66.54 \pm 0.84}$ |

The evaluation encompassed specific experimental parameters, including the number of rounds, model architecture, total number of clients, proportion of selected clients per dataset, batch size, and learning rate, as detailed in Table 1. For optimization, we employed an SGD optimizer with a momentum of 0.5 and a cross-entropy loss function on the client side. The data partitioning and noise model followed the approach in FedCorr (Xu et al., 2022), where $p \in (0, 1)$ denotes the class sampling probability of clients, $\rho \in [0, 1]$ denotes the proportion of noisy clients (with $\rho = 1$ indicating all clients are noisy), and $\tau \in [0, 1]$ represents the lower bound of the noise level for noisy clients.

### 4.2. Overall Comparison Results

We compared the predictive performance of FedClean with the aforementioned methods under label noise conditions, evaluating both IID and non-IID distributions on CIFAR-10, as well as IID distribution on CIFAR-100. The results are summarized in Tables 2 and 3, respectively. In both IID and non-IID settings, when all clients are noise-free, all methods exhibit similar strong performance. However, when noise is introduced, the performance of these methods diverges significantly. FedCorr performs well when not all clients are noisy, but when all clients are noisy, only our methods (FedClean[1] and FedClean[2]) maintain good performance. Other methods show a marked decline in performance under this condition. This supports our earlier claim that FedClean does not rely on the presence of clean clients, demonstrating robust performance even with a high proportion of noisy clients, as further discussed in Subsection 4.3. Additionally, although the performance of FedClean is not optimal when

*Table 4.* Accuracies (%) of various methods on Clothing1M with non-IID setting.

| Methods | FedAvg | FedProx | FedCorr | FedNoRo | FedBeat | FedELC | FedFixer | FedClean[1] | FedClean[2] |
|---|---|---|---|---|---|---|---|---|---|
| Acc | 68.63 | 69.15 | 69.02 | 69.21 | 67.05 | 69.24 | 70.52 | 70.17 | **72.39** |

not all clients are noisy, the difference between FedClean and the optimal performance is minimal. We attribute this slight performance gap to the use of relatively basic CNNL methods in our implementation. With more sophisticated CNNL approaches, we believe FedClean has significant potential for further improvement. We also conducted experiments in a non-IID setting using the real-world dataset Clothing1M. Notably, we did not introduce synthetic label noise, as the dataset already contains inherent tag noise, with all clients being noisy clients (i.e., $\rho = 1$). The experimental results, presented in Table 4, show that the proposed method FedClean[2] achieves the highest accuracy.

### 4.3. Robustness When All clients Are Noisy

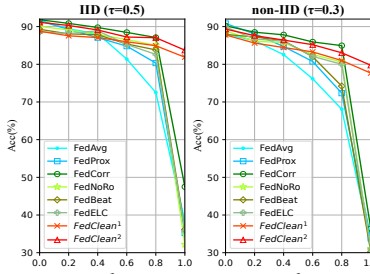

*Figure 2.* Accuracy (%) variations of various methods on the CIFAR-10 dataset as the ratio of noisy clients ($\rho$) increases under IID (with $\tau = 0.5$) and non-IID (with $\tau = 0.3$) settings.

Figure 2 illustrates the performance variations of various methods on the CIFAR-10 dataset as the ratio of noisy clients $\rho$ increases, with fixed lower bounds for the noise level under both IID and non-IID settings. The results show that when $\rho < 1$, all FNLL methods perform significantly better than the baseline, with FedCorr consistently achieving the best performance. However, when $\rho = 1$ (i.e., when all clients are noisy), only the FedClean methods maintains strong performance, while the performance of all other methods deteriorates sharply. Furthermore, the trend of performance decline reveals that the FedClean method exhibits the slowest decline, particularly when $\rho$ increases from 0.8 to 1. This highlights the robustness of FedClean, demonstrating its resilience against high ratio of noisy clients in FL, even in the absence of clean clients.

*Table 5.* Ablation study results (average and standard deviation of 5 trials) on CIFAR-10 of FedClean[2].

| Methods | IID | | non-IID | |
|---|---|---|---|---|
| | $\rho = 0.5$ | $\rho = 1$ | $\rho = 0.5$ | $\rho = 1$ |
| | $\tau = 0.3$ | $\tau = 0.5$ | $\tau = 0.3$ | $\tau = 0.5$ |
| Ours | **88.41** | **83.75** | **86.79** | **80.55** |
| Ours w/o CNLL | 77.52 | 47.74 | 71.30 | 40.11 |
| Ours w/o correction | 81.02 | 76.45 | 77.71 | 73.94 |
| Ours w/o correction I | 86.83 | 82.58 | 84.37 | 79.19 |
| Ours w/o correction II | 84.61 | 79.85 | 82.31 | 77.01 |
| Ours w/o ASSA | 87.77 | 82.91 | 85.98 | 79.28 |
| Ours w/o Mixup | 88.12 | 83.55 | 86.18 | 80.00 |

### 4.4. Ablation Study

Table 5 outlines the impact of the components in FedClean. We summarize key insights into FedClean's effectiveness: i) All components contribute to improved accuracy. ii) CNLL has the greatest influence. During the preprocessing phase, CNLL is essential for accurately selecting clean samples; only when clean samples are correctly identified can subsequent corrections be effective. iii) The robustness of FedClean to FL with a high proportion of noisy clients is primarily achieved through the local CNLL applied at clients.

## 5. Extension of ASSA

ASSA can be seamlessly extended to strengthen privacy protection. Inspired by ideas from zkCor (Wang et al., 2024), we employ the zero-knowledge proofs (ZKP) (Sun et al., 2021) for secure label noise correction in our FedClean. By requiring each client to provide a computation integrity proof, zkCor ensures the correctness of the label correction process while preserving privacy. This integration adds a layer of privacy protection, preventing malicious clients from affecting the global model with erroneous or tampered labels. Additionally, zkCor introduces a batch ZKP protocol that enhances verification efficiency. By allowing proofs to be verified in batches, we reduce the computational load on the aggregator, improving scalability. This batch verification is incorporated into ASSA, alleviating the aggregator's verification burden. zkCor is detailed in the Appendix D. (Notably, zkCor is an extension of ASSA designed to meet enhanced privacy requirements.)

## 6. Conclusion

In this paper, we introduced FedClean, a robust label noise correction method for FL that does not rely on the presence of clean clients. FedClean employs a two-stage framework to identify and correct noisy labels, leveraging a collaborative per-sample loss function to assess the confidence of label corrections. Additionally, we proposed ASSA to optimize the influence of each client during federated training, based on their contribution to the dataset size. Our experimental validation on the CIFAR-10/100 and Clothing1M datasets under both IID and non-IID conditions demonstrates the effectiveness of FedClean in mitigating label noise without compromising privacy.

While promising, our approach leaves room for exploration, particularly in refining the local CNLL model and addressing its complexity. Future work will optimize the CNLL process for clients and explore advanced techniques to enhance label noise correction in federated settings.

## Acknowledgments

This work was sponsored by the SEU Innovation Capability Enhancement Plan for Doctoral Students [CXJH SEU 24243], the National Natural Science Foundation of China [Grant 62076130], the Start-up Research Fund of Southeast University [Grant RF1028623059], and the Big Data Computing Center of Southeast University.

## Impact Statement

This paper presents work whose goal is to advance the field of Machine Learning. There are many potential societal consequences of our work, none which we feel must be specifically highlighted here.

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

# APPENDIX
# Supplemental Information for FedClean

## A. Why Choose Collaborative Per-sample Loss ?

We explain our choice to measure the confidence of noisy labels using cooperative per-sample loss, rather than relying solely on annotation label per-sample loss, from the following perspectives:

- *Noise in Annotation Labels:* If there is noise in the annotation label, $\mathcal{L}_{\text{an}}(y_k^i, \hat{y}_k^i; \theta_G^{T_1})$ will typically be large. The loss $\mathcal{L}_{\text{in}}(\bar{y}_k^i, \hat{y}_k^i; \theta_G^{T_1})$, which reflects the consistency of the inferred label with the true category of the sample, is generally smaller when the inferred labels are more consistent and accurate, as inferred labels are formed by grouping similar samples together.

- *Increase in Collaborative Loss:* If $\mathcal{L}_{\text{an}}(y_k^i, \hat{y}_k^i; \theta_G^{T_1})$ exceeds $\mathcal{L}_{\text{in}}(\bar{y}_k^i, \hat{y}_k^i; \theta_G^{T_1})$, the collaborative per-sample loss $\mathcal{L}_{\text{co}}(y_k^i, \bar{y}_k^i, \hat{y}_k^i; \theta_G^{T_1})$ will be larger, indicating that the annotation label is likely noisy. This increases the likelihood that the inferred label can provide a more accurate correction.

- *Correction Potential:* When the collaborative per-sample loss is large, the difference between the annotation label and inferred label is significant, suggesting that the sample can be corrected by the inferred label. Therefore, a larger collaborative per-sample loss implies a stronger correction effect of the inferred label on the sample.

## B. Extended Proof of Theorem 3.1

**Theorem 3.1.** *Incorporating inferred labels $\bar{y}_k^i$ as prior information refines the model's predictions, leading to a performance improvement bounded as:*

$$\mathcal{A}(\hat{y}_k^i, \bar{y}_k^i) \geq \mathcal{A}(\hat{y}_k^i) + \delta, \tag{19}$$

*where $\delta$ quantifies the reduction in prediction error after combining inferred labels.*

*Proof.* Let $(x_k^i, y_k^i, \bar{y}_k^i)$ be the $i$-th sample in the $k$-th client, where $x_k^i$ is the input feature, $y_k^i$ (true) is the true label, and $\bar{y}_k^i$ is the inferred label (or prior information). Let the global model $\hat{y}_k^i = \theta_G^{T_1}(x_k^i)$ represent the prediction made by the unrefined model.

We aim to refine the prediction by incorporating $\bar{y}_k^i$ as a prior. The process of updating the model's prediction based on this prior is grounded in Bayesian inference. Specifically, we update the posterior distribution of the model's prediction given the inferred label using Bayes' Theorem:

$$P(\hat{y}_k^i | \bar{y}_k^i) = \frac{P(\bar{y}_k^i | \hat{y}_k^i) \cdot P(\hat{y}_k^i)}{P(\bar{y}_k^i)}, \tag{20}$$

where:

- $P(\hat{y}_k^i | \bar{y}_k^i)$ is the posterior probability of the prediction given the inferred label;

- $P(\bar{y}_k^i | \hat{y}_k^i)$ is the likelihood of observing $\bar{y}_k^i$ given the prediction $\hat{y}_k^i$;

- $P(\hat{y}_k^i)$ is the prior probability of the prediction $\hat{y}_k^i$ from the global model;

- $P(\bar{y}_k^i)$ is the normalizing constant ensuring that the posterior sums to 1.

The update process refines the global model's prediction using the inferred labels as priors. The refined prediction is obtained by maximizing the posterior:

$$\hat{y}_k^i(\bar{y}_k^i) = \arg\max_{\hat{y}_k^i} P(\hat{y}_k^i|\bar{y}_k^i), \tag{21}$$

which is equivalent to MAP (performing Maximum A Posteriori) estimation.

To analyze the model's error, we define the error rate of the global model before incorporating the inferred labels as:

$$\epsilon = \frac{1}{S}\sum_{i=1}^{S}\mathbb{I}(\hat{y}_k^i \neq y_k^i(\text{true})), \tag{22}$$

where $\mathbb{I}(\cdot)$ is the indicator function and $S$ is the total number of samples.

After incorporating the inferred labels as priors, the error rate becomes:

$$\epsilon' = \frac{1}{S}\sum_{i=1}^{S}\mathbb{I}(\hat{y}_k^i(\bar{y}_k^i) \neq y_k^i(\text{true})), \tag{23}$$

where $\hat{y}_k^i(\bar{y}_k^i)$ is the refined prediction after updating with the inferred label $\bar{y}_k^i$.

Now, let us express the accuracy of the model before and after incorporating the inferred labels. The accuracy without the prior is:

$$\mathcal{A}(\hat{y}_k^i) = 1 - \epsilon, \tag{24}$$

and the accuracy after incorporating the inferred labels is:

$$\mathcal{A}(\hat{y}_k^i, \bar{y}_k^i) = 1 - \epsilon'. \tag{25}$$

Thus, the improvement in accuracy is:

$$\delta = \epsilon - \epsilon' = \frac{1}{S}\sum_{i=1}^{S}\left[\mathbb{I}(\hat{y}_k^i \neq y_k^i(\text{true})) - \mathbb{I}(\hat{y}_k^i(\bar{y}_k^i) \neq y_k^i(\text{true}))\right]. \tag{26}$$

We are guaranteed that $\delta \geq 0$, since incorporating prior information cannot worsen the model's prediction.

In order to quantify the reduction in prediction error more rigorously, we can relate the error reduction to the KL (Kullback-Leibler) divergence between the model's prior and posterior distributions.

The KL divergence between the prior and posterior distributions of the prediction $\hat{y}_k^i$ given the inferred label $\bar{y}_k^i$ is defined as:

$$D_{\text{KL}}(P(\hat{y}_k^i|\bar{y}_k^i)||P(\hat{y}_k^i)) = \mathbb{E}_{P(\hat{y}_k^i|\bar{y}_k^i)}\left[\log\frac{P(\hat{y}_k^i|\bar{y}_k^i)}{P(\hat{y}_k^i)}\right]. \tag{27}$$

This measures the informational difference between the prior and posterior distributions. A smaller KL divergence indicates that incorporating the inferred labels has effectively refined the model's predictions.

Thus, we can use the following bound on the improvement in accuracy:

$$\delta \approx -\frac{1}{S}\sum_{i=1}^{S}D_{\text{KL}}(P(\hat{y}_k^i|\bar{y}_k^i)||P(\hat{y}_k^i)), \tag{28}$$

where the approximation assumes that the reduction in error is closely related to the KL divergence between the prior and posterior distributions.

This formulation provides a more nuanced view of the improvement in prediction accuracy by leveraging the Bayesian framework. Specifically, the reduction in error is not just a simple difference in prediction success, but is linked to the information gain obtained from the inferred labels.

Therefore, incorporating inferred labels $\bar{y}_k^i$ as prior information refines the global model's predictions and leads to an improvement in accuracy bounded as:

$$\mathcal{A}(\hat{y}_k^i, \bar{y}_k^i) \geq \mathcal{A}(\hat{y}_k^i) + \delta, \quad \delta = \epsilon - \epsilon' \quad \text{and} \quad \delta \approx -\frac{1}{S} \sum_{i=1}^{S} D_{\mathrm{KL}}(P(\hat{y}_k^i | \bar{y}_k^i) || P(\hat{y}_k^i)). \tag{29}$$

The accuracy improvement is hence driven by the Bayesian updating process, and the reduction in prediction error is quantitatively related to the information gain through the KL divergence between the prior and posterior distributions. $\quad\square$

## C. Extended Proof of Theorem 3.2

**Theorem 3.2.** *In collaborative per-sample losses, inferred labels provide additional stability, allowing for more accurate estimates of true labels.*

*Proof.* We aim to compute the posterior distribution $P(y_k^i | \hat{y}_k^i, \bar{y}_k^i)$, which represents the belief about the annotation label $y_k^i$ given the model's prediction $\hat{y}_k^i$ and the inferred label $\bar{y}_k^i$.

By Bayes' theorem, we can express the posterior distribution as:

$$P(y_k^i | \hat{y}_k^i, \bar{y}_k^i) \propto P(\hat{y}_k^i | y_k^i) P(\bar{y}_k^i | y_k^i) P(y_k^i), \tag{30}$$

where:

- $P(\hat{y}_k^i | y_k^i)$ is the likelihood of observing the model's prediction given the annotation label;

- $P(\bar{y}_k^i | y_k^i)$ is the likelihood of observing the inferred label given the annotation label;

- $P(y_k^i)$ is the prior.

Now, let us analyze how the inferred labels provide stability. We introduce the Bayesian evidence in the form of the marginal likelihood of the inferred label $\bar{y}_k^i$ by marginalizing out $y_k^i$:

$$P(\bar{y}_k^i | \hat{y}_k^i) = \sum_{y_k^i} P(\bar{y}_k^i | y_k^i) P(y_k^i | \hat{y}_k^i), \tag{31}$$

which can be interpreted as the expected likelihood of $\bar{y}_k^i$, accounting for all possible true labels weighted by their posterior probability given $\hat{y}_k^i$. The inferred label $\bar{y}_k^i$ plays a crucial role in stabilizing the model's estimates, particularly in scenarios where annotations are noisy. When the likelihood $P(\hat{y}_k^i | y_k^i)$ is low due to noise, the presence of a reliable inferred label $\bar{y}_k^i$ can compensate for this uncertainty and improve the overall estimate quality.

To quantify this stability, we introduce the expected log-likelihood as a measure of model uncertainty:

$$\mathcal{L}_{\text{stability}} = \mathbb{E}_{P(y_k^i | \hat{y}_k^i, \bar{y}_k^i)}[\log P(y_k^i | \hat{y}_k^i, \bar{y}_k^i)]. \tag{32}$$

This quantity reflects the model's confidence in its estimates of $y_k^i$, and a higher value indicates greater stability. When $P(\hat{y}_k^i | y_k^i)$ is low (due to noisy annotations), the inferred label $\bar{y}_k^i$ increases the likelihood of the annotation label $y_k^i$, thus increasing the stability of the model's predictions.

We can also formalize the stability improvement using the KL divergence, which measures the difference between the posterior distribution and the prior distribution:

$$D_{\mathrm{KL}}(P(y_k^i | \hat{y}_k^i, \bar{y}_k^i) || P(y_k^i)) = \mathbb{E}_{P(y_k^i | \hat{y}_k^i, \bar{y}_k^i)} \left[ \log \frac{P(y_k^i | \hat{y}_k^i, \bar{y}_k^i)}{P(y_k^i)} \right]. \tag{33}$$

The reduction in KL divergence after incorporating $\bar{y}_k^i$ reflects the improvement in the model's stability and accuracy. A smaller KL divergence implies that the model's posterior has become closer to the true distribution of $y_k^i$, due to the stabilizing effect of $\bar{y}_k^i$.

---

*Protocol 1 (zkCor\*):* Let  be the security parameter.

- **Step 1:** Given the security parameter λ, the public parameters *pp* are generated.

- **Step 2:** For each client $k$, the private dataset $\mathcal{D}_k$ is committed using the Dataset Commitment protocol described in Subsection D.1.

- **Step 3:**
    (1) Each client performs FedClean on dataset $\mathcal{D}_k$ to generate the arithmetic circuit $\mathcal{C}$ and corresponding wire values as witnesses $w_k$.
    (2) Each client performs Proof Generation (Subsection D.3) with the private dataset $\mathcal{D}_k$, witnesses $w_k$, arithmetic circuit $\mathcal{C}$, and public parameter *pp*, producing the proof $\pi$ and the *value* needed in ASSA.

- **Step 4:** The aggregator performs Batch Verification (Subsection D.3) using the proofs $\pi_k$, commitments $cm_k$, arithmetic circuit $\mathcal{C}$, *value* , and public parameter pp. If the verification succeeds, it outputs 1 and accepts the *value*; otherwise, it rejects the results and aborts.

---

*Figure 3.* The zkCor\* protocol for FedClean.

Thus, we can conclude that incorporating inferred labels into the model's decision-making process leads to an improvement in stability, especially when annotations are noisy. This results in a more accurate estimate of the annotation label $y_k^i$, as the inferred label $\bar{y}_k^i$ provides additional information that counteracts the effects of noise.

$$P(y_k^i|\hat{y}_k^i, \bar{y}_k^i) \geq P(y_k^i|\hat{y}_k^i), \quad \text{where} \quad \delta = D_{\text{KL}}(P(y_k^i|\hat{y}_k^i, \bar{y}_k^i)||P(y_k^i)) \geq 0. \tag{34}$$

$\square$

## D. zkCor in ASSA

We incorporate zkCor into our ASSA, and since we modified the arithmetic circuit in zkCor to align with our ASSA algorithm, the resulting protocol, referred to as zkCor\*, is illustrated in Figure 3.

### D.1. Dataset Commitment

For each client $k \in \mathcal{N}$, it commits to the private dataset $D_k$ using the randomness $r_k$ and the public parameter $pp$, generating the commitment $cm_k$ (Kate et al., 2010). Specifically, for the item $a_{xy}^{(k)}$ in the $x$-th row and $y$-th column,

$$f_{\mathcal{D}_k}(z) = a_{xy}^{(k)} \quad \text{where} \quad z = n(x-1) + y, \tag{35}$$

where $n$ is the number of columns in the dataset, the client $k$ applies the Kate polynomial commitment scheme to generate $cm_k$.

### D.2. Noise Correction

Our proposed FedClean noise correction scheme requires clients to upload the number of local samples involved in model updates during each training iteration. The aggregator then updates the global model using the ASSA aggregation scheme, while clients must prove the computational integrity of their local label noise correction. To accomplish this, both the client and aggregator generate three arithmetic circuits $\mathcal{C}_{(1)}, \mathcal{C}_{(2)}, \mathcal{C}_{(3)}$ for calculating the number of locally selected samples in Eqs (4), (15), and (18) as public information. FedClean can calculate the dataset sizes for the three ASSAs as follows:

$$|\mathcal{D}_k^c| = \sum_{i=1}^{n_k} \mathbb{I}(y_k^i = \bar{y}_k^i), \tag{36}$$

$$|\widetilde{\mathcal{D}_k^n}'| = \sum_{i=1}^{n_k} \mathbb{I}(y_k^i = \bar{y}_k^i) - |\mathcal{D}_k^c|, \tag{37}$$

$$|\widehat{\mathcal{D}_k^{n}}'| + |\mathcal{D}_k^{n_2}| = n_k - \sum_{i=1}^{n_k} \mathbb{I}(y_k^i \neq \hat{y}_k^i) - |\mathcal{D}_k^c| - |\widetilde{\mathcal{D}_k^{n}}'|. \tag{38}$$

In Eq 38, $\hat{y}_k^i$ represents the prediction label for the global model $\theta_G^{T_1+T_2}$.

To verify the integrity of each client's local computation in each iteration, it suffices to validate the correctness of the values $|\mathcal{D}_k^c|$, $|\widetilde{\mathcal{D}_k^{n}}'|$, and $|\widehat{\mathcal{D}_k^{n}}'| + |\mathcal{D}_k^{n_2}|$ (which are updated after each local processing.) by converting Eqs (36), (37), and (38) as follows:

$$|\mathcal{D}_k^c| + \left\{ -\left[ \sum_{i=1}^{n_k} \mathbb{I}(y_k^i = \bar{y}_k^i) \right] \right\} = 0, \tag{39}$$

$$|\widetilde{\mathcal{D}_k^{n}}'| + \left\{ -\left[ \sum_{i=1}^{n_k} \mathbb{I}(y_k^i = \bar{y}_k^i) - |\mathcal{D}_k^c| \right] \right\} = 0, \tag{40}$$

$$|\widehat{\mathcal{D}_k^{n}}'| + |\mathcal{D}_k^{n_2}| + \left\{ -\left[ n_k - \sum_{i=1}^{n_k} \mathbb{I}(y_k^i \neq \hat{y}_k^i) - |\mathcal{D}_k^c| - |\widetilde{\mathcal{D}_k^{n}}'| \right] \right\} = 0. \tag{41}$$

The above arithmetic can be expressed in the ZKP arithmetic circuit. To facilitate the operation of the counter $\mathbb{I}(\cdot)$, we define the $n_k$-dimensional vectors $I_k^{(1)}$ and $I_k^{(2)}$:

$$
\begin{aligned}
I_k^{(1)}[i] &= \begin{cases} 1 & \text{if } y_k^i = \bar{y}_k^i, \\ 0 & \text{if } y_k^i \neq \bar{y}_k^i, \end{cases} \quad 0 \leq i < n_k. \\[2ex]
I_k^{(2)}[i] &= \begin{cases} 1 & \text{if } y_k^i = \hat{y}_k^i, \\ 0 & \text{if } y_k^i \neq \hat{y}_k^i, \end{cases} \quad 0 \leq i < n_k,
\end{aligned}
\tag{42}
$$

Using the arithmetic circuits $\mathcal{C}_k^{(1)}, \mathcal{C}_k^{(2)}, \mathcal{C}_k^{(3)}$ computed from $|\mathcal{D}_k^c|$, $|\widetilde{\mathcal{D}_k^{n}}'|$, and $|\widehat{\mathcal{D}_k^{n}}'| + |\mathcal{D}_k^{n_2}|$, the client could generate the proofs $\pi_k^{(t)}$, as described in the following subsection.

### D.3. Proof Generation

In the proposed FedClean scenario, proofs from each client use the same arithmetic circuits $\mathcal{C}_{(1)}, \mathcal{C}_{(2)}, \mathcal{C}_{(3)}$, derived from subsection D.2. To prevent the aggregator from repeatedly verifying each proof, a new batch ZKP protocol is introduced in zkCor, allowing the aggregator to verify all proofs in a single process.

**High-level Idea of Batch ZKP.** Let $\left\{ \mathcal{P}_k^{(t)} \right\}_{k=1}^q$ denote the set of clients (provers) participating in the round $t$ parameter update of FedClean, where $q = Fraction \times N$ (with Fraction representing the proportion of clients involved in FL training). In the FedClean algorithm, the $\mathcal{C}_{(1)}$ arithmetic circuit is used for $1 \leq t \leq T_1$, the $\mathcal{C}_{(2)}$ circuit for $T_1 + 1 \leq t < T_1 + T_2$, and the $\mathcal{C}_{(3)}$ circuit for $T_1 + T_2 + 1 \leq t \leq T_1 + T_2 + T_3$. The batch ZKP protocol requires a distinct Lagrange basis set for $\left\{ \mathcal{P}_k^{(t)} \right\}_{k=1}^q$ when interpolating polynomials. Assuming a subgroup $H$ of order $q^n$, the highest polynomial order in the ZKP protocol is $q^n - 1$. $\mathcal{P}_k^{(t)} \in \left\{ \mathcal{P}_k^{(t)} \right\}_{k=1}^q$ is restricted to ues $\left( L_{(k-1) \times q^{n-1}+1}^{(t)}, L_{(k-1) \times q^{n-1}+2}^{(t)}, ..., L_{k \times q^{n-1}}^{(t)} \right)$, where $L_i$ is the $i$-th Lagrange base.

Suppose $\left\{ \mathcal{P}_k^{(t)} \right\}_{k=1}^q$ have witnesses vectors $\left\{ \mathbf{a}_k^{(t)} \right\}_{k=1}^q$. Their corresponding interpolation polynomials are $\left\{ f_k^{(t)}(X) \right\}_{k=1}^q$. By applying Kate polynomial commitment (Kate et al., 2010), there are commitments $\left\{ \left[ f_k^{(t)}(X) \right]_1 \right\}_{k=1}^q$. The verifier (aggregator, the server in FedClean.) is able to compute

$$\left[ f^{(t)}(X) \right]_1 = \sum_{k=1}^q \left[ f_k^{(t)}(X) \right]_1. \tag{43}$$

Let the vector that is corresponding to the polynomial $f^{(t)}(X)$ be $\mathbf{a}$, which is the concatenation of $\left\{ \mathbf{a}_k^{(t)} \right\}_{k=1}^q$. For any random challenge $r \in \mathbb{F}$ ($\mathbb{F}$ is the finite field.) chosen by the verifier, $\left\{ \mathcal{P}_k^{(t)} \right\}_{k=1}^q$ provide their openings at the random point

$r$ as $\left\{ f_k^{(t)}(r) \right\}_{k=1}^{q}$:

$$f^{(t)}(r) = \sum_{k=1}^{q} f_k^{(t)}(r),$$

$$h^{(t)}(X) = \sum_{k=1}^{q} h_k^{(t)}(X),$$

(44)

where $\left\{ h_k^{(t)}(X) \right\}_{k=1}^{q}$ and $h^{(t)}(X)$ are defined as

$$h_k^{(t)}(X) = \frac{f_k^{(t)}(X) - f_k^{(t)}(r)}{X - r}, \quad k = 1, ..., q,$$

$$h^{(t)}(X) = \frac{f^{(t)}(X) - f^{(t)}(r)}{X - r}.$$

(45)

As long as $\left\{ \mathcal{P}_k^{(t)} \right\}_{k=1}^{q}$ follow the protocol and compute their polynomials using the correct Lagrange basis, their witnesses can be combined without introducing new random numbers, allowing the verifier to validate them together. To ensure proper use of the specified Lagrange basis, the verifier defines the vector $\left\{ \mathbf{s}_k^{(t)} \right\}_{k=1}^{q}$, where $\mathbf{s}_k^{(t)} = (0, ..., 0, 1..., 1, 0, ..., 0)$ with $q^{n-1}$ ones spanning form $\mathbf{s}_k^{(t)}[(k-1) \times q^{n-1} + 1]$ to $\mathbf{s}_k^{(t)}[k \times q^{n-1}]$ and the $(q-1) \times q^{n-1}$ zeros for the rest. The verifier would require all $\left\{ \mathcal{P}_k^{(t)} \right\}_{k=1}^{q}$ to demonstrate

$$f_k^{(t)}(X) = S_k^{(t)}(x) \cdot f_k^{(t)}(X), \quad k = 1, ..., q,$$

(46)

where $\left\{ S_k^{(t)}(X) \right\}_{k=1}^{q}$ is the degree $q^n$ polynomial corresponding to $\left\{ \mathbf{s}_k^{(t)} \right\}_{k=1}^{q}$, where $X \in [1, q^n]$. Eq (46) is called the interpolation constraints.

**Algorithm Description.** All $q$ provers preserves the wire values of the arithmetic circuits $\mathcal{C}_{(1)}, \mathcal{C}_{(2)}, \mathcal{C}_{(3)}$ as witnesses. Specifically, suppose the provers $\left\{ \mathcal{P}_k^{(t)} \right\}_{k=1}^{q}$ have the witnesses $\left\{ \left( \mathbf{a}_k^{(t)}, \mathbf{b}_k^{(t)}, \mathbf{c}_k^{(t)} \right) \right\}_{k=1}^{q}$. The provers first rearrange their witnesses as $(w_{ki})_{i=1}^{3\eta}$, which is the concatenated witnesses of $\mathbf{a}_k^{(t)}$, $\mathbf{b}_k^{(t)}$ and $\mathbf{c}_k^{(t)}$.

Suppose the order of subgroup $H$ is $q^n$. The degrees of all the polynomials in the Plonk protocol are lower than $m$ ($m < q^{(n-1)}$). The parties agree that $\left\{ \mathcal{P}_k^{(t)} \right\}_{k=1}^{q}$ use non-coincident Lagrange interpolation bases. Besides, all provers will add the randomnesses when constructing the polynomials to guarantee the property for zero-knowledge:

$$a_k^{(t)}(X) = (b_{k1}X + b_{k2}) \cdot Z_H(X) + \sum_{i=1}^{q^{n-1}} w_{ki} L_{i+(k-1) \times q^{n-1}}(X),$$

$$b_k^{(t)}(X) = (b_{k3}X + b_{k4}) \cdot Z_H(X) + \sum_{i=1}^{q^{n-1}} w_{k(\eta+i)} L_{i+(k-1) \times q^{n-1}}(X),$$

(47)

$$c_k^{(t)}(X) = (b_{k5}X + b_{k6}) \cdot Z_H(X) + \sum_{i=1}^{q^{n-1}} w_{k(2\eta+i)} L_{i+(k-1) \times q^{n-1}}(X),$$

where $w_{ki}, i \in [1, 6]$ are the random numbers chosen by $\mathcal{P}_k^{(t)}$. These randomness are used to mask the secret polynomials of each prover. $Z_H(X)$ is the zero polynomial, which is defined as $Z_H(X) = X^{\eta} - 1$. And $H$ is the multiplicative subgroup with $\omega$ as the $\eta$-th root of unity and the generator of the subgroup, such that $H = \{1, \omega, ..., \omega^{\eta-1}\}$.

The verifier chooses $\beta, \gamma \xleftarrow{\$} \mathbb{F}_p$ to all of the provers ($\beta, \gamma \xleftarrow{\$} \mathbb{F}_p$ denotes random sampling, where $\alpha$ and $\beta$ are uniformly

selected from the finite field $\mathbb{F}_p$ of prime order $p$.), who will compute the permutation polynomial $\left\{z_k^{(t)}(X)\right\}_{k=1}^q$ as follows:

$$
\begin{aligned}
z_k^{(t)}(X) = & (b_{k7}X^2 + b_{k8}X + b_{k9}) \cdot Z_H(X) + L_{1+(k-1)\times q^{n-1}}(X) \\
& + \sum_{i=1}^{\eta-1}\left(L_{i+1+(k-1)\times q^{n-1}}(X)\prod_{j=1}^{i}\prod_{l=0}^{2}\frac{\mathrm{w}_{k(\eta l+j)}+\beta\xi_l\omega^{j-1}+\gamma}{\mathrm{w}_{k(\eta l+j)}+\beta\sigma(\eta l+j)+\gamma}\right), \qquad k=1,...,q,
\end{aligned}
\tag{48}
$$

where $\xi_0$ is defined to be 0 and $b_{kj}, k \in [1,q], j \in [7,9]$ are randomnesses chosen by $\mathcal{P}_k^{(t)}$. $\left\{z_k^{(t)}(X)\right\}_{k=1}^q$ are represented as permutation polynomials to encode permutation constraints in a thPlonk constraint system. $\sigma(\cdot)$ is a permutation defined on the field $\mathbb{H}'$ of size $[3\eta]$, such that $\sigma : [3\eta] \to [3\eta]$. The field $\mathbb{H}'$ is defined as $\mathbb{H}' := \mathbb{H} \cup (\xi_1 \cdot \mathbb{H}) \cup (\xi_2 \cdot \mathbb{H})$, where $\xi_1, \xi_2 \in \mathbb{F}$ are chosen such that $\mathbb{H}, \xi_1 \cdot \mathbb{H}$, and $\xi_2 \cdot \mathbb{H}$ are distinct cosets of $\mathbb{H}$, ensuring $\mathbb{H}'$ contains $3\eta$ distinct elements.

After computing the permutation polynomials, $\left\{\mathcal{P}_k^{(t)}\right\}_{k=1}^q$ assign them individually. The verifier selects $\alpha \overset{\$}{\leftarrow} \mathbb{F}_p$ and sends it to the provers, who then compute the quotient polynomials $\left\{t_k^{(t)}(X)\right\}_{k=1}^q$:

$$
\begin{aligned}
t_k^{(t)}(X) = & \frac{1}{Z_H(X)} \cdot \Big((z_k^{(t)}(X)-1)L_{1+(k-1)\times q^{n-1}}(X)\alpha^2 + a_k^{(t)}(X)(1-S_k^{(t)}(X))\alpha^3 \\
& + a_k^{(t)}(X)b_k^{(t)}(X)q_M(X) + a_k^{(t)}(X)q_L(X) + b_k^{(t)}(X)q_R(X) + c_k^{(t)}(X)q_O(X) \\
& + \alpha(a_k^{(t)}(X)+\beta X+\gamma)(c_k^{(t)}(X)+\beta\xi_2 X+\gamma)(a_k^{(t)}(X)+\beta X+\gamma)z_k^{(t)}(X) \\
& - \alpha z_k^{(t)}(X\omega)(a_k^{(t)}(X)+\beta S_{\sigma 1}(X)+\gamma)(b_k^{(t)}(X)+\beta S_{\sigma 2}(X)+\gamma) \\
& \cdot (c_k^{(t)}(X)+\beta S_{\sigma 3}(X)+\gamma) + PI(X) + q_C(X)), \qquad k=1,...,q,
\end{aligned}
\tag{49}
$$

$\mathcal{P}_k^{(t)}$ splits the quotient polynomial $t_k^{(t)}(X)$ into two polynomials, $t_{i\_lo}^{(t)}(X)$ and $t_{i\_mi}^{(t)}(X)$, each of degree less than $\eta$, and a polynomial $t_{i\_hi}^{(t)}(X)$ of degree at most $\eta + 5$:

$$
t_k^{(t)}(X) = t_{k\_lo}^{(t)'}(X) + X^\eta \cdot t_{k\_mid}^{(t)'}(X) + X^{2\eta} \cdot t_{k\_hi}^{(t)'}(X), \quad k=1,...q.
\tag{50}
$$

The provers mask these polynomials with random integers $b_{k10}, b_{k11} \in \mathbb{F}_p$ and define:

$$
\begin{aligned}
t_{k\_lo}^{(t)}(X) :&= t_{k\_lo}^{(t)'}(X) + b_{k10}X^\eta, \\
t_{k\_mi}^{(t)}(X) :&= t_{k\_mi}^{(t)'}(X) - b_{k10} + b_{k11}X^\eta, \\
t_{k\_hi}^{(t)}(X) :&= t_{k\_mi}^{(t)'}(X) - b_{k11}, \quad k=1,...,q.
\end{aligned}
\tag{51}
$$

Note that:

$$
t_k^{(t)}(X) = t_{k\_lo}^{(t)}(X) + X^\eta \cdot t_{k\_mi}^{(t)}(X) + X^{2\eta} \cdot t_{k\_hi}^{(t)}(X).
\tag{52}
$$

The verifier selects the evaluation challenge $z \overset{\$}{\leftarrow} \mathbb{F}_p$ and sends it to the provers, who computes the open evaluations $\bar{a}_k, \bar{b}_k$, $\bar{c}_k, \bar{s}_{\sigma_1 k}, \bar{s}_{\sigma_2 k}, \bar{t}_k, \bar{z}_{k\omega}$, and the linearized polynomial $r_k^{(t)}(X)$, such that:

$$
\begin{aligned}
r_k^{(t)}(X) = & \bar{a}_k(1-S_k^{(t)}(X))\alpha^3 + z_k^{(t)}(X)L_\mu(z)\alpha^2 \\
& + ((\bar{a}_k+\beta_k z+\gamma_k)(\bar{b}_k+\beta_k\xi_1 z+\gamma_k)(\bar{c}_k+\beta_k k_2 z+\gamma_i)z_i(X))\alpha \\
& - ((\bar{a}_i+\beta_i\bar{s}_{\sigma_1 k}+\gamma_i)(\bar{b}_i+\beta_i\bar{s}_{\sigma_2 k}+\gamma_k)\beta_k z_k(z\omega) \cdot S_{\sigma_3}(X))\alpha \\
& + \bar{a}_k\bar{b}_k q_M(X) + \bar{a}_k q_L(X) + \bar{b}_k q_R(X) + \bar{c}_k q_O(X) + q_c(X), k=1,...,q.
\end{aligned}
\tag{53}
$$

All of the provers compute the openings of $\left\{r_k^{(t)}(X)\right\}_{k=1}^q$ at the evaluation challenge point $z$.

The verifier selects challenge $z \xleftarrow{\$} \mathbb{F}_p$ and sends it to the provers, who can calculate the opening polynomials $W_{kz}^{(t)}(X)$, $W_{k(z\omega)}^{(t)}(X)$ and commit to them:

$$W_{kz}^{(t)}(X) = \frac{1}{X-z} \cdot \begin{pmatrix} r_k^{(t)}(X) + v(a_k^{(t)}(X) - \bar{a}_k) + v^2(b_k^{(t)}(X) - \bar{b}_k) \\ +v^3(c_k^{(t)}(X) - \bar{c}_k) + v^4(S_{\sigma_1}(X) - \bar{s}_{\sigma_1 k}) \\ +v^5(S_{\sigma_2(X)} - \bar{s}_{\sigma_2 k}), \end{pmatrix} \tag{54}$$

$$W_{k(z\omega)}^{(t)}(X) = \frac{z_k^{(t)}(X) - \bar{z}_{k\omega}}{X - z\omega}, \quad k = 1, ..., q. \tag{55}$$

The commitments to the above polynomials are denoted as $\left[W_{kz}^{(t)}\right]_1 := \left[W_{kz}^{(t)}(X)\right]_1$ and $\left[W_{k(z\omega)}^{(t)}\right]_1 := \left[W_{k(z\omega)}^{(t)}(X)\right]_1$.

Finally, the verifier will receive proofs $\left\{\pi_k^{(t)}\right\}_{k=1}^q$:

$$\pi_i^{(t)} = \begin{pmatrix} \left[a_k^{(t)}\right]_1, \left[b_k^{(t)}\right]_1, \left[c_k^{(t)}\right]_1, \left[z_k^{(t)}\right]_1, \left[t_{k\_lo}^{(t)}\right]_1, \left[t_{k\_mi}^{(t)}\right]_1, \left[t_{k\_hi}^{(t)}\right]_1 \\ \left[W_{kz}^{(t)}\right]_1, \left[W_{k(z\omega)}^{(t)}\right]_1, \bar{a}_k, \bar{b}_k, \bar{c}_k, \bar{s}_{\sigma_1 k}, \bar{s}_{\sigma_2 k}, \bar{z}_{k\omega} \end{pmatrix}, \quad k = 1, ..., q. \tag{56}$$

## D.4. Batch Verification

The verifier waits for all proofs to be provided and then performs batch validation. Specifically, the verifier first preprocesses all public information defined by the arithmetic circuit, which is independent of both the witnesses and the public input $w$. Additionally, the verifier computes polynomial commitments for the interpolation constraints to ensure that each prover uses a distinct set of Lagrange bases:

$$\left[1 - S_k^{(t)}\right]_1 := \left(1 - S_k^{(t)}(X)\right) \cdot [1]_1. \tag{57}$$

Upon receiving proofs $\left\{\pi_k^{(t)}\right\}_{k=1}^q$ from each prover, the verifier combines all witnesses and then verifies the correctness of the proofs:

$$\left[a^{(t)}\right]_1 = \sum_{k=1}^q \left[a_k^{(t)}\right]_1,$$

$$\left[b^{(t)}\right]_1 = \sum_{k=1}^q \left[b_k^{(t)}\right]_1,$$

$$\left[c^{(t)}\right]_1 = \sum_{k=1}^q \left[c_k^{(t)}\right]_1,$$

$$\left[z^{(t)}\right]_1 = \sum_{k=1}^q \left[z_k^{(t)}\right]_1,$$

$$\left[t_{lo}^{(t)}\right]_1 = \sum_{k=1}^q \left[t_{k_{lo}}^{(t)}\right]_1, \tag{58}$$

$$\left[t_{mi}^{(t)}\right]_1 = \sum_{k=1}^q \left[t_{k_{mi}}^{(t)}\right]_1,$$

$$\left[t_{hi}^{(t)}\right]_1 = \sum_{k=1}^q \left[t_{k_{hi}}^{(t)}\right]_1,$$

$$\left[W_z^{(t)}(X)\right]_1 = \sum_{k=1}^q \left[W_{kz}^{(t)}(X)\right]_1,$$

$$\left[W_{z\omega}^{(t)}(X)\right]_1 = \sum_{k=1}^q \left[W_{k(z\omega)}^{(t)}(X)\right]_1, \quad k = 1, ..., q.$$

Besides, the verifier combines the evaluation values:

$$
\begin{aligned}
\bar{a} &= \sum_{k=1}^{q} \bar{a}_k, \\
\bar{b} &= \sum_{k=1}^{q} \bar{b}_k, \\
\bar{c} &= \sum_{k=1}^{q} \bar{c}_k, \\
\bar{r} &= \sum_{k=1}^{q} \bar{r}_k, \\
\bar{t} &= \sum_{k=1}^{q} \bar{t}_k, \\
\bar{z}\omega &= \sum_{k=1}^{q} \bar{z}\omega_k, \quad k = 1, ..., q.
\end{aligned}
\tag{59}
$$

Then the verifier computes the partial opening commitment $[D]_1$:

$$
\left[ D^{(t)} \right]_1 := v \cdot \sum_{k=1}^{q} \left[ r_k^{(t)} \right]_1 + u \cdot \sum_{k=1}^{q} \left[ z_k^{(t)} \right]_1 .
\tag{60}
$$

Note that in Eq (60), there are commitments for interpolating constrained polynomials, integrated by powers of a random challenge $\alpha$ sent by the verifier. Based on the opening commitment, the verifier then computes the batch polynomial commitment $\left[ F^{(t)} \right]_1$ and the batch evaluation $\left[ E^{(t)} \right]_1$:

$$
\begin{aligned}
\left[ F^{(t)} \right]_1 &:= \left[ D^{(t)} \right]_1 + v \cdot \left[ a^{(t)} \right]_1 + v^2 \cdot \left[ b^{(t)} \right]_1 + v^3 \cdot \left[ c^{(t)} \right]_1 + v^4 \cdot [s_{\sigma_1}]_1 + v^5 \cdot [s_{\sigma_2}]_1 \\
\left[ E^{(t)} \right]_1 &:= (-r_0 + v\bar{a} + v^2\bar{b} + v^3\bar{c} + v^4\bar{s}_{\sigma_1} + v^5\bar{s}_{\sigma_2} + u\bar{z}_\omega) \cdot [1]_1 .
\end{aligned}
\tag{61}
$$

Finally, the verifier checks the equality of the following bilinear paring:

$$
e\left( \left[ W_z^{(t)} \right]_1 + u \cdot \left[ W_{z\omega}^{(t)} \right]_1 , [x]_2 \right) \overset{?}{=} e\left( z \left[ W_z^{(t)} \right]_1 + uz\omega \left[ W_{z\omega}^{(t)} \right]_1 + \left[ F^{(t)} \right]_1 - \left[ E^{(t)} \right]_1 , [1]_2 \right) .
\tag{62}
$$

If Eq (62) holds, the verifier accepts all batch proofs from the provers; otherwise, the verifier rejects the protocol and aborts.

## D.5. Security Analysis

**Theorem D.1.** *The zkCor\* scheme, as described in Figure 3 and Subsections D.1, D.2, D.3 and D.4, is a zero-knowledge scheme for FedClean.*

*Proof.* To demonstrate that zkCor\* is zero-knowledge for FedClean, we address three key properties: completeness, soundness, and zero-knowledge.

**Completeness.** The verification process outputs 1 if the $|\mathcal{D}_k^c|$, $|\widetilde{\mathcal{D}_k^n}'|$, and $|\widehat{\mathcal{D}_k^n}'| + |\mathcal{D}_k^{n_2}|$ are correctly computed by the clients, as per Eqs (39) (40) (41), and their committed datasets. The completeness of zkCor\* follows from the correctness of the batch ZKP protocol we propose, which is straightforward to validate.

**Soundness.** Let $\mathcal{C} = \{\mathcal{C}_{(1)}, \mathcal{C}_{(2)}, \mathcal{C}_{(3)}\}$ represent the arithmetic circuit defined in Subsections D.1, D.2, D.3 and D.4. By the extractability of the polynomial commitment scheme used in the batch ZKP protocol, there exists an extractor $\varphi$ that, given the commitment $cm$, extracts a witness $w^* = (D^*, aux)$ such that $cm = $ zkCor\*.Com$(D^*, r, pp)$ with overwhelming probability, where $aux$ represents the auxiliary witnesses generated during computation. If $cm = $ zkCor\*.Com$(D^*, r, pp)$ and zkCor\*.$\mathcal{V}(\pi, cm, \mathcal{C}, value, pp) = 1$ ($value = |\mathcal{D}_k^c|$, $|\widetilde{\mathcal{D}_k^n}'|$, $|\widehat{\mathcal{D}_k^n}'| + |\mathcal{D}_k^{n_2}|$ reps. $\mathcal{C} = \mathcal{C}_{(1)}, \mathcal{C}_{(2)}, \mathcal{C}_{(3)}$.), but $value$ is incorrect, two scenarios can arise:

- Scenario 1: $w = (D^*, aux)$ satisfies $\mathcal{C}$. There are three possibilities:

  (i) $D^*$ is not the committed dataset but passes the commitment verification. This is negligible in $\lambda$ due to the soundness of the polynomial commitment scheme (Kate et al., 2010).

  (ii) $value$ is incorrect but passes the batch ZKP verification. This is negligible in $\lambda$ due to the soundness of the ZKP scheme. The concatenation of commitments and allocation of distinct Lagrange bases does not compromise soundness (Gabizon et al., 2019).

  (iii) Some witnesses in $aux$ are incorrect but satisfy the circuits $\mathcal{C}$. This is negligible in $\lambda$ for the same reason as in (ii).

- Scenario 2: $w = (D, aux)$ does not satisfy $\mathcal{C}$. The soundness of the proposed batch ZKP ensures that the probability of a client generating a simulated proof that causes the aggregator to accept is negligible in $\lambda$.

**Zero-Knowledge.**    The zero-knowledge property follows from the characteristics of PLONK (Gabizon et al., 2019). The distribution of the Lagrange bases does not affect the original zero-knowledge property in the Plonk protocol.    □

