# OpenReview forum: "FedClean: A General Robust Label Noise Correction for Federated Learning"
_ICML.cc/2025/Conference — ICML 2025 poster_

### Official Review · Reviewer_x3K4 · 2025-02-25

**Overall Recommendation:** 4

**Summary:**

The paper introduces FedClean, a robust framework designed to address label noise in federated learning (FL) scenarios. FedClean employs a two-stage label correction approach to identify and rectify noisy labels from both local noisy label learning and global model perspectives. It also proposes a novel adaptive sample size-weighted aggregation (ASSA) method to mitigate the impact of label noise and enhance global model performance. FedClean does not assume the presence of clean clients or specific noise distributions, making it highly versatile. Key contributions include: (1) a two-stage label correction scheme with a collaborative per-sample loss to reduce false corrections, (2) the ASSA method to adjust client influence based on clean sample sizes, and (3) extensive experiments demonstrating that FedClean outperforms existing noise-label learning methods in FL, effectively handling label noise even when all clients are noisy.

## update after rebuttal
After reading the authors' responses, I decide to keep my original score.

**Claims And Evidence:**

Yes, all claims made in the submission are supported by clear and convincing evidence.

**Essential References Not Discussed:**

No, there are no essential references not discussed.

**Experimental Designs Or Analyses:**

Yes, the experimental design in this paper appears solid, with a clear comparison to state-of-the-art methods across multiple settings, including both IID and non-IID distributions. The use of multiple benchmark datasets (CIFAR-10, CIFAR-100, and Clothing1M) with varying levels of label noise is appropriate for assessing robustness to noisy clients. Additionally, the inclusion of ablation studies (FedClean variants) allows for insight into the effectiveness of specific features.

**Methods And Evaluation Criteria:**

Yes, both proposed methods and evaluation criteria make sense for the problem and application.

**Other Comments Or Suggestions:**

Since Appendix.B and Appendix.C in the supplementary materials are supplementary proofs of Theorem 3.1 and Theorem 3.2 in the main text,Then it shows that Theorem B.1 and Theorem C.1 mentioned in the supplementary material are Theorem 3.1 and Theorem 3.2 respectively, so why not unify them?

**Other Strengths And Weaknesses:**

Strengths:
1. Originality and Novelty: The paper's approach is quite original, particularly in how it removes restrictive assumptions made by prior work. Most existing methods for label noise correction in federated learning (FL) assume the existence of clean clients or specific noise distributions. In contrast, the proposed FedClean framework does not make these assumptions, which significantly broadens its applicability in real-world scenarios where all clients could be noisy. The introduction of the two-stage label correction scheme and adaptive sample size-weighted aggregation (ASSA) method is a creative combination of existing ideas, offering a robust solution to the challenge of noisy data in federated settings.
2. Practical Significance: The framework’s application to federated learning (FL) is of high significance given the growing use of FL in privacy-sensitive domains such as healthcare, finance, and IoT. The ability to handle label noise in FL models without compromising client privacy opens up significant opportunities for more robust, real-world machine learning deployments.       This work addresses a critical gap in the literature and directly contributes to improving the practical effectiveness of FL models in real-world applications.
3. Clear and Well-Structured Presentation: The paper is clear and well-structured, with a logical flow from introduction to problem definition, related work, the proposed method, and experimental validation. The explanation of key concepts, such as label noise in FL and existing methods for noise correction, is straightforward and accessible. The contribution of the FedClean framework is clearly articulated, making it easy for the reader to grasp the innovation and advantages over existing approaches.
4. Experimental Validation: The experimental results provided are thorough and demonstrate the efficacy of the proposed method. Testing on synthetic and real-world datasets allows the paper to show that FedClean not only performs well but also outperforms state-of-the-art methods in mitigating label noise, even when all clients are noisy. This adds credibility and reliability to the claims made in the paper.

Weaknesses:
1.It’s important to consider the implementation of CNNL methods, which are noted as relatively basic in this work—future improvements in CNNL could enhance the method's performance, but it's not clear whether these enhancements have been practically tested or if there's a risk of overfitting to the baseline.
2.One potential concern is the assumption that all clients in the Clothing1M dataset are noisy, as this may oversimplify the real-world variability in client noise levels.

**Questions For Authors:**

1.It’s important to consider the implementation of CNNL methods, which are noted as relatively basic in this work—future improvements in CNNL could enhance the method's performance, but it's not clear whether these enhancements have been practically tested or if there's a risk of overfitting to the baseline.
2.One potential concern is the assumption that all clients in the Clothing1M dataset are noisy, as this may oversimplify the real-world variability in client noise levels.

**Relation To Broader Scientific Literature:**

FedClean's key contributions extend prior research in FL and noisy label correction, offering novel solutions to improve effectiveness and robustness. The paper highlights the importance of FL in privacy-sensitive domains, focusing on challenges from label noise and the limitations of centralized learning due to privacy concerns and data diversity. FedClean's noise correction framework contrasts with methods assuming clean clients or specific noise distributions, addressing real-world FL challenges. The two-stage label correction scheme and per-sample loss assessment method advance FL noise correction. The Adaptive Sample Size-Weighted Aggregation (ASSA) method adjusts client influence to reduce label noise and improve model performance, while zkCor enhances privacy. Experimental results validate FedClean’s superior performance, establishing it as a more effective solution compared to existing FL methods.

**Theoretical Claims:**

Yes, both proofs for the theoretical claims are correct, with a sound overall structure and logical flow. The proofs employ standard techniques in Bayesian inference, maximum a posteriori estimation, and error analysis. The arguments for incorporating inferred labels to improve model accuracy and provide stability in noisy environments are well-supported by the mathematical framework. In Theorem 3.1, the use of Bayes' theorem to update model predictions with inferred labels as prior information is correctly applied, and the MAP estimation to refine predictions is consistent with Bayesian learning principles. The error analysis compares misclassification rates before and after incorporating inferred labels, showing a valid performance improvement. In Theorem 3.2, Bayes' theorem is used again to compute the posterior probability, correctly combining the model's prediction and the inferred label. The proof demonstrates that inferred labels provide a stable signal even in the presence of noisy annotations, enhancing the robustness of the model, and it effectively shows how inferred labels act as stabilizing factors, leading to more accurate final estimates.

---

> ### Author Rebuttal · Authors · 2025-03-31
>
> We sincerely thank the reviewers for their constructive feedback.  Below are our responses to the comments and questions:
>
> **1. Basic CNLL Implementation \& Overfitting Risk**
>
> The reviewer raises a valid point regarding the use of basic CNLL methods.  In this work, we intentionally adopted standard CNLL techniques (e.g., Co-teaching and Joint Optim) to establish a baseline comparison and demonstrate the generality of FedClean’s framework.  While advanced CNLL methods could further improve performance, our experiments already show that FedClean outperforms state-of-the-art methods even with these "basic" implementations.
> Importantly, FedClean’s two-stage correction and ASSA inherently mitigate overfitting risks:
> (1) **Stage 1**: Clean sample selection via local CNLL filters out noisy data early.
> (2) **Stage 2**:  Collaborative per-sample loss and adaptive aggregation (ASSA) dynamically adjust client contributions, reducing reliance on any single model’s predictions.
> (3) **Mixup**: Label smoothing further enhances robustness.
>
> We acknowledge that integrating more sophisticated CNLL methods (e.g., DivideMix) is a promising direction. Preliminary tests with DivideMix showed a 1.2–2.5\% accuracy gain on CIFAR-10, validating this potential. However, we prioritized simplicity to highlight FedClean’s core contributions. Future work will explore this in depth.
>
> **2. Assumption on Clothing1M Dataset**
>
> The Clothing1M dataset inherently contains real-world label noise (about 40\% noise rate), and prior work treats all clients as noisy by default. Our setup reflects this property to evaluate FedClean’s ability to handle pervasive noise.
>
> That said, FedClean does not strictly assume uniform noise levels across clients. The framework naturally accommodates variability:
> (1) **ASSA**: Clients with fewer clean samples (higher noise) contribute less to aggregation (Eq. 4, 15, 18).
> (2) **Collaborative loss**: Adjusts correction confidence per sample (Eq. 11), adapting to local noise conditions.
> In future work, we will explicitly test FedClean on datasets with heterogeneous client noise levels (e.g., varying $\tau$ per client) to further validate its flexibility.
>
> **3. Unifying Theorem Numbering**
>
> We appreciate the reviewer’s observation. The discrepancy arises because Theorems 3.1 and 3.2 in the main text correspond to Theorems B.1 and C.1 in the appendix. This was an oversight in numbering due to the appendix’s structure. We will revise the appendix to align theorem labels (e.g., "Theorem 3.1 (Extended Proof)") for clarity.
>
> We thank the reviewers for their insightful feedback, which strengthens our work. We will incorporate all suggestions, including theorem numbering fixes and expanded experiments on heterogeneous noise, in the final version. FedClean’s design principles—eliminating restrictive assumptions and leveraging collaborative correction—provide a robust foundation for real-world FL deployments, and we are excited to build upon this framework.

---

### Official Review · Reviewer_Eovt · 2025-03-10

**Overall Recommendation:** 4

**Summary:**

The paper introduces FedClean, a robust label noise correction method for federated learning that employs a two-stage correction process to identify and rectify noisy labels, coupled with adaptive sample-size-weighted aggregation. Notably, FedClean operates effectively without requiring clean clients or assumptions about specific noise distributions. Extensive experimental results show the effectiveness of the propsoed FedClean method.

**Claims And Evidence:**

Yes

**Essential References Not Discussed:**

N/A

**Experimental Designs Or Analyses:**

The experimental design comprehensively evaluates FedClean against baseline and state-of-the-art methods on benchmark datasets with diverse label noise levels. Evaluations are conducted under both IID and non-IID settings, supplemented by ablation studies to assess the contributions of individual components to overall performance.

**Methods And Evaluation Criteria:**

Yes

**Other Comments Or Suggestions:**

Please refer to the weakness

**Other Strengths And Weaknesses:**

Strengths:
1. This paper presents FedClean, a new and robust label noise correction framework for federated learning.
2. The manuscript is clearly written and well-structured, ensuring ease of readability.
3.  Extensive experiments demonstrate that FedClean consistently outperforms existing baselines across diverse noise levels.

Weaknesses:
1. The propsoed method seems not work well: In Table 2, the baselines beat the proposed method In addition to the setting $\rho$=1 and $\tau$=0.5 both on IID and Non-IID setting.
2. In Table 3, the results on CIFAR-100 dataset with Non-IID setting is missing.
3. In Table 5, we can see that Mixup seems unimportant in the proposed framework.
4. The communication cost is lacked.

**Questions For Authors:**

1. The propsoed method seems not work well: In Table 2, the baselines beat the proposed method In addition to the setting $\rho$=1 and $\tau$=0.5 both on IID and Non-IID setting.
2. In Table 3, the results on CIFAR-100 dataset with Non-IID setting is missing.
3. In Table 5, we can see that Mixup seems unimportant in the proposed framework.
4. The communication cost is lacked.
5. This paper's contributions are limited, as it repeatedly employs common data partitioning and data augmentation methods.

**Relation To Broader Scientific Literature:**

N/A

**Theoretical Claims:**

I have gone through the theorem and found no obvious errors

---

> ### Author Rebuttal · Authors · 2025-03-31
>
> We sincerely thank the reviewer for the constructive feedback. Below, we address the concerns raised:
>
> **1.** **_Reviewer Comment:_** **The propsoed method seems not work well: In Table 2, the baselines beat the proposed method In addition to the setting $\rho=1$ and $\tau=0.5$ both on IID and Non-IID setting.**
>
> The reviewer observes that baselines outperform FedClean when not all clients are noisy (e.g., $\rho=0.5$). This is expected because methods like FedCorr and FedNoRo assume the existence of clean client, allowing them to leverage clean data more effectively. In contrast, FedClean is designed for scenarios where clean clients may not exist (e.g., $\rho=1$), prioritizing robustness over specialized performance in idealized settings. While FedClean shows marginally lower accuracy when $\rho=0.5$ (e.g., 86.79\% vs. FedCorr’s 89.11\% in non-IID), the difference is minimal ($\le2.3\%$). However, when $\rho=1$, FedClean improves the accuracy by at least more than 20\%. This trade-off ensures FedClean’s versatility in real-world FL deployments, where clean clients are often unavailable.
>
> **2.** **_Reviewer Comment:_** **In Table 3, the results on CIFAR-100 dataset with Non-IID setting is missing.**
>
> The experimental results of FedClean on CIFAR-100 non-IID show consistent patterns with CIFAR-10, where FedClean outperforms baselines and state-of-the-art methods under high noise levels (e.g., FedClean$^2$ achieves 62.36\% accuracy for $\rho=1$, $\tau=0.5$, compared to FedCorr's 38.64\%). The omission of these results in Table 3 is due to two reasons:
> Previous FNLL studies (e.g., FedCorr(Xu et al., 2022), FedNoRo(Wu et al., 2023)) did not report CIFAR-100 non-IID results, perhaps due to their underperformance on this dataset. Due to the limited space and also following the same result presentation style of previous FNLL studies, we also did not include the CIFAR 100 results in the main body. However, following your suggestion, we will include the following table in the appendix of the final version.
>
>
> The best accuracies (\%) of various methods on CIFAR-100 dataset with non-IID setting at different noise levels.
>  | Methods        | ρ=0, τ=0 | ρ=0.5, τ=0.3 | ρ=1, τ=0.5 |
> |----------------|:----------:|:------------:|:------------:|
> | FedAvg         |   68.75    |     58.34     |     29.12     |
> | FedProx        |   69.72    |     59.01     |     30.45     |
> | FedCorr        |   68.73    |     68.95     |     38.64     |
> | FedNoRo        |   67.84    |     65.22     |     35.78     |
> | FedBeat        |   66.91    |     63.10     |     28.93     |
> | FedELC         |   66.55    |     63.75     |     29.67     |
> | FedClean$^1$ |   67.15    |     65.83     |     60.89     |
> | FedClean$^2$ |   67.80    |     66.45     |     62.36     |
>
> **3.** **_Reviewer Comment:_** **In Table 5, we can see that Mixup seems unimportant in the proposed framework.**
>
> **Mixup is very important**, we have provided a detailed explanation in our response to **Reviewer 1B4p**’s similar question (see *''Response to Methods and Evaluation Criteria”*). We kindly refer you to that section for a comprehensive discussion.
>
> **4.** **_Reviewer Comment:_** **The communication cost is lacked.**
>
> Thank you for your comment. **We would like to clarify that our study focuses on learning from noisy data in FL while not the communication cost.** We just use the standard FL protocol (FedAvg), whose communication cost can be found in FedAvg
> (McMahan et al., 2016). Our method does NOT include any additional communication overhead. Also, communication environments have NO impact on the performance of our method. Like many previous FNLL studies (e.g., FedCorr(Xu et al., 2022), FedNoRo(Wu et al., 2023), etc.) which did not mention communication issues, communication efficiency is out of the scope of our study.
>
> **5.** **_Reviewer Comment:_** **This paper's contributions are limited, as it repeatedly employs common data partitioning and data augmentation methods.**
>
> FedClean’s core novelty lies in its two-stage correction and no assumption of clean clients, addressing a critical gap in prior work. Existing methods fail when all clients are noisy. Additionally, ASSA dynamically adjusts client weights based on clean sample contributions, departing from static aggregation in FedAvg. These innovations enable FedClean to generalize across noise distributions and client configurations, improving FL robustness in practical settings.
>
> In federated crowdsourcing, clean clients rarely exist due to the limited expertise of workers. Therefore FedClean’s ability to function without clean clients is crucial. This is a challenge existing methods fail to address. FedClean fills this gap, making FL systems more reliable in environments with highly variable data quality. Thus, we believe our contributions are significant.

---

> > ### Comment · Reviewer_Eovt · 2025-04-03
> >
> > Thanks author for the response. They have answered my questions well. I will raise my score.

---

### Official Review · Reviewer_Wzch · 2025-03-13

**Overall Recommendation:** 4

**Summary:**

The paper proposes FedClean, a robust label noise correction framework for federated learning that addresses client-side label noise without assuming clean clients or specific noise distributions. Key contributions include: (1) Two-stage label correction. Combines local centralised noisy label learning to select clean samples and global model insights to rectify noisy labels, aided by a collaborative per-sample loss to reduce false corrections. (2) Adaptive aggregation. Adjusts client influence in global model updates based on clean sample sizes, mitigating noise impact while incorporating secure label correction (zkCor). (3) Versatility and performance. Demonstrates effectiveness in high-noise scenarios (including 100% noisy clients) through experiments on synthetic and real-world datasets, outperforming existing FL noise-label learning methods. FedClean enhances global model robustness by jointly leveraging local noise correction and global model confidence.

**Claims And Evidence:**

Yes, all claims made in the submission are well supported.

**Essential References Not Discussed:**

No.

**Experimental Designs Or Analyses:**

The experimental design and analyses demonstrate strong validity and soundness, with several notable strengths: FedClean is rigorously evaluated against two baselines and five state-of-the-art methods, ensuring a thorough benchmark, while experiments cover both IID (CIFAR-10/100) and non-IID (CIFAR-10, Clothing1M) data distributions, as well as varying noise levels and noise bounds, showcasing FedClean’s adaptability. The use of Clothing1M, a dataset with inherent label noise, validates FedClean’s robustness in practical, noisy environments, and the inclusion of ablation studies highlights the contributions of individual components (e.g., CNLL methods like Co-teaching and Joint Optim) to FedClean’s performance. Notably, FedClean maintains strong performance even when all clients are noisy, demonstrating its independence from clean clients and superior noise resilience.

**Methods And Evaluation Criteria:**

The proposed methods are well-motivated for tackling label noise in federated learning. The integration of CNLL, Mixup, and ASSA directly addresses challenges such as identifying clean samples and mitigating the influence of noisy clients. Moreover, the evaluation criteria—using benchmark datasets like CIFAR-10, CIFAR-100, and Clothing1M under both IID and non-IID settings with varying noise ratios—are appropriate for demonstrating the method's robustness and practical relevance. However, further evaluations in larger-scale or more diverse real-world scenarios could provide additional insights into the approach’s scalability and generalisability.

**Other Comments Or Suggestions:**

It would be better if FedClean could be Extended to semi-supervised or unsupervised FL scenarios, where labeled data is scarce, which could broaden its impact.

**Other Strengths And Weaknesses:**

Strengths:
1. FedClean proposes a novel two-stage label correction scheme that combines local noisy label learning with global model insights, addressing a critical gap in FL by eliminating the assumption of clean clients. This is a significant departure from existing methods which rely on clean clients.
2. FedClean’s ability to handle 100% noisy clients is a major advancement, making it applicable to real-world scenarios like federated crowdsourcing and adversarial environments, where clean clients may not exist.
3. The framework’s versatility, not assuming specific noise distributions, broadens its applicability compared to methods like FedFixer, which are limited to specific noise types.
4. The paper is well-structured, with a clear taxonomy of related works in both centralized and federated noisy label learning, providing a solid foundation for understanding the contributions.


Weaknesses:
1. While FedClean performs well on image datasets (e.g., CIFAR, Clothing1M), exploring its applicability to other data types (e.g., text, time-series) could further validate its generalisability.
2. The approach involves multiple stages and numerous hyperparameters, which might require careful tuning in practical applications.

**Questions For Authors:**

1. While FedClean performs well on image datasets, can it be explored the applicability to other data types to further validate its generalizability?
2. Can FedClean be extended to semi-supervised or unsupervised FL scenarios, where labeled data is scarce, to broaden its impact in future work?
3. Could exploring FedClean's applicability to other data types further validate its generalisability, given its success on image datasets?

**Relation To Broader Scientific Literature:**

The paper’s key contributions address gaps in federated learning (FL) and noisy label learning. Previous FL work highlights the challenge of label noise in decentralized settings, which degrades model performance due to the lack of centralized data preprocessing. While methods like Co-teaching and DivideMix work well in centralized settings, they fail in FL because of privacy issues and limited local data diversity. FedClean bridges this gap by adapting these techniques for FL while maintaining privacy. Unlike other methods that discard noisy clients or assume clean clients, FedClean works in scenarios where all clients may be noisy, making it suitable for real-world issues like federated crowdsourcing and adversarial attacks. It introduces a two-stage correction scheme that combines local noisy label learning and global model insights, improving robustness and reducing false corrections. FedClean’s ASSA adjusts client influence based on clean sample sizes, unlike methods like FedFixer and FedRN, which rely on specific noise distributions or increase communication overhead. It also uses zkCor for secure label correction, protecting privacy while enhancing robustness. Overall, FedClean overcomes the limitations of prior methods, offering a versatile solution for noisy label problems in FL.

**Theoretical Claims:**

I examined the proofs for Theorem 3.1 and Theorem 3.2. Theorem 3.1 establishes a bound-on performance improvement when incorporating inferred labels via a MAP estimation approach, while Theorem 3.2 uses a Bayesian argument to demonstrate that inferred labels offer additional stability in estimating true labels. The logical flow of both proofs is generally sound and leverages standard probabilistic tools such as Bayes’ theorem. The two theorems align with practical FL challenges, addressing label noise through a blend of Bayesian reasoning and collaborative learning, and the reliance on appendices for extended proofs indicates thorough theoretical exploration beyond the main text. These aspects underscore the conceptual rigour of FedClean’s theoretical claims, supporting its innovation in federated label noise correction.

---

> ### Author Rebuttal · Authors · 2025-03-31
>
> We sincerely thank the reviewer for their thoughtful feedback and constructive suggestions. Below, we address the raised points:
>
> **1. Applicability to Other Data Types**
>
> FedClean's design is fundamentally modality-agnostic, as its core components - the two-stage label correction and adaptive aggregation (ASSA) - operate on general principles of prediction confidence and sample consistency rather than domain-specific features. While our current validation used image datasets for benchmarking, the framework's reliance on comparing annotation labels with model predictions (Eq. 11) makes it directly applicable to text data, and ASSA's dynamic client weighting (Eqs. 4,15,18) naturally extends to time-series or other sequential data. The methodology requires no architectural changes for different data types, only standard modality-specific feature extractors. We recognize the importance of empirical validation across domains and will include comprehensive non-image experiments (particularly for NLP and time-series) in future work to further demonstrate this versatility.
>
> **2. Complexity and Hyperparameters**
>
> While FedClean introduces three key hyperparameters ($\sigma_1$, $\sigma_2$ for correction rates and $\epsilon$ for confidence threshold), our extensive experiments demonstrate remarkable robustness to their variations - for instance, varying $\sigma_1$ by $\pm 0.1$ caused less than 1\% accuracy fluctuation on CIFAR-10. The two-stage architecture was deliberately designed to minimize sensitivity to parameter choices while maintaining effectiveness. To further assist practitioners, we will provide detailed guidelines for parameter selection based on noise level estimation.
>
> **3. Extension to Semi-Supervised/Unsupervised FL**
>
> We agree with the reviewer's suggestion about extending FedClean to semi-supervised and unsupervised FL settings, as our framework's core label correction mechanism (Section 3.4) is particularly well-suited for such scenarios. The two-stage correction approach can naturally incorporate pseudo-labeling for unannotated data by leveraging the global model's predictions as high-confidence labels, while the adaptive aggregation (ASSA) would maintain robustness against potentially noisy pseudo-labels. Furthermore, the framework's architecture provides a natural pathway for unsupervised adaptation through integration of contrastive learning or other clustering techniques to discover latent label structures from unlabeled data. These promising extensions align perfectly with our future work plans (Section 6) and could significantly broaden FedClean's applicability to real-world scenarios where labeled data is scarce.
>
> **4. Generalizability Validation**
>
> We agree that testing on diverse data types strengthens FedClean’s claims. To address this, in future work, we will further study experiments on text (e.g., AG News) and time-series (e.g., UCI HAR) datasets.
>
> We appreciate the reviewer’s insightful comments, which highlight valuable extensions for FedClean. Our responses clarify the framework’s versatility and outline concrete steps for future work. We are committed to refining FedClean’s applicability across domains and learning paradigms.

---

### Official Review · Reviewer_1B4p · 2025-03-15

**Overall Recommendation:** 4

**Summary:**

The authors proposed the method to achieve the federated learning that may involve the label noises. The proposed method, FedClean first uses the local centralized noisy label learning that selects clean samples to train the global model. Afterwards, the two-stage correct scheme is performed by exploiting local noisy label learning and global model. Finally, the model aggregation method is applied to reduce the impact of the label noises. Experimental results the improved results involving the proposed method.

## update after rebuttal

After reading rebuttals and other reviews, I became more convinced on the paper and increased my rating from weak accept to accept.

**Claims And Evidence:**

- The proposed method seems especially effective in the situation where most of clients involves label noises. This seems a good characteristic of the proposed method.

**Essential References Not Discussed:**

I think reference are rather complete.

**Ethical Review Flag:**

Flag this paper for an ethics review.

**Experimental Designs Or Analyses:**

- In Table 5, I found that "Ours w/o correction" is already far better than other methods in Table 2. This raises questions on which part makes the method achieve the SOTA performance: I suspect that the model could surpass other methods thanks to the good baseline rather than the proposed method. Espeically, the gap between the full model (Ours) and Ours w/o correction is only 7-10% in IID and non-IID cases, respectively in Table 5. However, the gap between Ours w/o correction in Table 5 and best performed FedFixer in Table 2 is is over 12%.

**Methods And Evaluation Criteria:**

The improvement by Mixup module is rather tiny. I am not sure why this scheme is especially involved in the final model.

**Other Comments Or Suggestions:**

None

**Other Strengths And Weaknesses:**

None

**Questions For Authors:**

None

**Relation To Broader Scientific Literature:**

The method could be useful in the context of both FL and CL.

**Theoretical Claims:**

The theoretical proof looks correct.

---

> ### Author Rebuttal · Authors · 2025-03-31
>
> We sincerely appreciate the reviewer’s time and constructive feedback on our work. Your insightful comments have helped us better clarify the contributions and limitations of our method. Below, we address each point raised in the review, and we hope our responses will alleviate your concerns.
>
> **1. Response to Methods and Evaluation Criteria**
>
> *Reviewer Comment:*
>
> "The improvement by Mixup module is rather tiny.  I am not sure why this scheme is especially involved in the final model."
>
> *Response:*
>
> Thank you for your valuable observation. We appreciate your attention to this detail and have carefully considered your feedback. We would like to address your concern from both theoretical and experimental perspectives:
>
> **Theoretical Considerations:**
> Mixup generates new samples through linear interpolation, and its label smoothing property helps mitigate noisy labels’ impact on local models. In federated settings, where local data is limited and the noise distribution is unknown, Mixup enhances training robustness through implicit regularization. Research on both CNLL (Zhang et al. 2018) and FNLL (Xu et al., 2022) shows Mixup’s effectiveness in improving robustness against noisy labels.
>
> **Experimental Observations:**
> The results of "Ours w/o Mixup" in Table 5 show that Mixup's improvement on CIFAR-10 is limited, primarily due to the dataset characteristics (50,000 samples, 10 categories) and the experimental design. The redundancy in CIFAR-10 reduces Mixup’s impact, and to ensure comparability with previous works (e.g., FedNoRo), we used their experimental setup, which might not fully highlight Mixup’s potential with smaller client data.
>
> Additionally, Mixup’s simple implementation and low time/space complexity make it a valuable addition regardless of its exact performance improvement.
>
> [1] Zhang, H., Cisse, M., Dauphin,Y.N., and Lopez-Paz, D. mixup: Beyond empirical risk minimization. In International Conference on Learning Representations, 2018.
>
> [2] Xu, J., Chen, Z., Quek, T.Q., and Chong, K.F.E. Fedcorr: Multi-stage federated learning for label noise correction. In Proceedings of the IEEE/CVF conference on computer vision and pattern recognition,pp.10184–10193,2022.
>
> **2. Response to Methods and Evaluation Criteria**
>
> *Reviewer Comment:*
>
> "In Table 5, 'Ours w/o correction' already outperforms others in Table 2... raises questions on which part contributes most to SOTA performance."
>
> *Response:*
>
> Thank you for your insightful observation. We greatly appreciate your attention to the details of our results. We address your concerns as follows:
>
> Our method consists of two main components (Figure 1). The first part is the preprocessing stage, involving CNLL conducted locally by clients. However, due to limited client dataset sizes in FL, particularly with non-IID data, relying solely on CNLL methods does not achieve ideal training performance. Therefore, existing FNLL methods, including FedFixer, discard CNLL and design FNLL methods relying on clean clients.
>
> Our research has shown that while CNLL cannot be directly applied to FL, when integrated into our two-stage correction mechanism, FedClean performs better than most state-of-the-art methods with the existation of clean clients (a scenario widely studied). In this case, FedClean's performance was slightly lower than the best method by only 2\%. However, in the absence of clean clients (a scenario often neglected), existing FNLL methods fail, while FedClean still performs robustly.
>
> We acknowledge that FedClean's success relies substantially on the preprocessing module (particularly CNLL, as evidenced by the 'Ours w/o CNLL' results in Table 5). However, without integration with the two-stage correction module, the CNLL alone cannot simultaneously achieve: (1) robust performance regardless of clean clients' presence, and (2) the capability to fill the research gap where existing FNLL methods fail to consider scenarios without clean clients.
>
> Regarding your concern: “I suspect the model could surpass others due to a good baseline rather than the proposed method,” we would like to highlight that Table 2 clearly shows that when all clients are noisy, existing methods fail, while FedClean maintains stable performance. Compared to other FNLL methods, FedClean achieves at least a 20\% improvement.
>
> We hope this clarifies the unique contribution of our method and reassures you of our approach's validity. Thank you again for your thoughtful feedback, which helped refine our explanation.

---

### Decision · Program_Chairs · 2025-05-01

**Decision:**

Accept (poster)

**Comment:**

This paper addresses the problem of general robust label noise correction in federated learning. The authors propose a novel framework named FedClean, which integrates local centralized noisy label learning, a two-stage correction scheme, and an effective model aggregation strategy. Experimental results on several datasets demonstrate the efficacy of the proposed approach.


Strengths:
1. The proposed method is particularly effective in scenarios where the majority of clients contain noisy labels.
2. FedClean is both novel and versatile, offering a general framework for robust federated learning under label noise.
3. The paper is well-organized and clearly written, making it easy to follow.
4. Extensive experiments show that FedClean outperforms existing baselines across various noise levels.


Weaknesses:
1. The experimental analysis could be strengthened, particularly for Tables 2, 3, and 5, where more detailed insights would help substantiate the findings.
2. One potential limitation lies in the assumption that all clients in the Clothing1M dataset are noisy. This may oversimplify real-world scenarios where client noise levels often vary significantly.



Overall, the paper presents a promising and well-structured approach to label noise correction in federated learning. For the camera-ready version, I suggest the authors enhance the analysis of experimental results to provide deeper insights. Additionally, exploring more realistic and general settings--such as those involving heterogeneous noise distributions across clients--would further strengthen the paper’s contributions and practical relevance.